# Stable water isotopes in the MITgcm

Rike Völpel [1], André Paul[1], Annegret Krandick[1], Stefan Mulitza[1] and Michael Schulz[1]

[1]MARUM - Center for Marine Environmental Sciences and Faculty of Geosciences, University of Bremen, Bremen, Germany

*Correspondence to*: Rike Völpel (rvoelpel@marum.de)

**Abstract.**

We present the first results of the implementation of stable water isotopes in the ocean general circulation model MITgcm. The model is forced with the isotopic content of precipitation and water vapor from an atmospheric general circulation model (NCAR IsoCAM), while the fractionation during evaporation is treated explicitly in the MITgcm. Results of the equilibrium simulation under pre-industrial conditions are compared to observational data and measurements of plankton tow records (the oxygen isotopic composition of planktic foraminiferal calcite). The broad patterns and magnitude of the stable water isotopes in annual mean seawater are well captured in the model, both at the sea surface as well as in the deep ocean. However, the surface water in the Arctic Ocean is not depleted enough, due to the absence of highly depleted precipitation and snow fall. A model-data mismatch is also recognizable in the isotopic composition of the seawater-salinity relationship in mid-latitudes that is mainly caused by the coarse grid resolution. Deep-ocean characteristics of the vertical water mass distribution in the Atlantic Ocean closely resemble observational data. The reconstructed $\delta^{18}O_c$ at the sea surface shows a good agreement with measurements. However, the model-data fit is weaker when individual species are considered and deviations are most likely attributable to the habitat depth of the foraminifera. Overall, the newly developed stable water isotope package opens wide prospects for long-term simulations in a paleoclimatic context.

## 1 Introduction

Stable water isotopes ($H_2^{16}O$, $H_2^{18}O$ and $HD^{16}O = HDO$) are widely used tracers of the hydrological cycle (Craig and Gordon, 1965; Gat and Gonfiantini, 1981) and can be used to determine the origin and mixing pattern of different water masses (e.g. Jacobs et al., 1985; Khatwala et al., 1999; Meredith et al., 1999). Due to differences in their physical and chemical properties, stable water isotopes undergo fractionation processes at any phase transition within the hydrological cycle (Craig and Gordon, 1965). This leads to distinctive isotopic signatures for different freshwater fluxes, which are commonly expressed as $\delta i$ (i = $^{18}O$ or D) with reference to the Vienna Standard Mean Ocean Water (VSMOW) standard and given as:

$$\delta i = \left( \frac{R}{R_{VSMOW}} - 1 \right) \cdot 1000 \text{ \textperthousand} , \tag{1}$$

where R is the ratio of the abundance of the heavier water isotope $H_2^{18}O$ or HDO to the abundance of the lighter isotope $H_2^{16}O$ and $R_{VSMOW} = 2005.2 \cdot 10^{-6}$ for $\delta^{18}O$ (Baertschi, 1976) and $155.95 \cdot 10^{-6}$ for $\delta D$ (de Wit et al., 1980).

Stable water isotopes have been used as an important proxy in a wide range of climate archives, e.g. in polar ice cores which provide past temperature records reflecting climatic changes over the past glacial-interglacial cycles (e.g. Dansgaard et al., 1969; Epstein et al., 1970; Johnsen et al., 1972; Johnsen et al., 2001) as well as speleothems which reveal intensity changes and variations in the amount of monsoonal rainfall (e.g. Wang et al., 2001; Fleitmann et al., 2003). As an indirect record, stable

water isotopes are preserved in carbonates ($CaCO_3$) from marine species such as planktonic and benthic foraminifers. Due to the temperature-dependent fractionation effect that occurs during the formation of $CaCO_3$, the oxygen isotopic composition of foraminiferal $CaCO_3$ ($\delta^{18}O_c$) is a function of both the ambient temperature and the isotopic composition of the seawater ($\delta^{18}O_w$) in which the calcification takes place (Emiliani, 1955). Hence, $\delta^{18}O_c$ records from sediment cores provide information on water mass changes.

During the last few decades, stable water isotopes have been incorporated more extensively in general circulation models (GCMs), first in atmospheric GCMs (AGCMs – e.g. Joussaume et al., 1984; Jouzel et al., 1987) and more than a decade later in oceanic GCMs (OGCMs – e.g. Schmidt, 1998; Paul et al., 1999; Delaygue et al., 2000; Wadley et al., 2002; Roche et al., 2004; Xu et al., 2012). In OGCMs the focus was mainly on the relationship between $\delta^{18}O_w$ and salinity, which are affected by similar physical processes. This topic is of significant interest in paleoceanography, because it is likely that changes in

advection and freshwater budgets as well as the source of precipitation may have altered this relationship (Rohling and Bigg, 1998). Using real freshwater flux boundary conditions in conjunction with the nonlinear free-surface (Huang, 1993) is essential to simulate it properly. Together, they ensure a dynamically more accurate simulation of the salinity due to the concentration and dilution effect and thus a freely evolving salinity at the sea surface. The Massachusetts Institute of Technology general circulation model (MITgcm) offers this very opportunity and further provides the adjoint method to perform data assimilation

(Errico, 1997).

Here, we present first results of the implementation of stable water isotopes in the MITgcm, by performing an equilibrium pre-industrial (PI) simulation and comparing it to available observations and reconstructions.

## 2 Methods

### 2.1 Ocean Model

We used the MITgcm "checkpoint" 64w, which refers to a specific time and/or point within the development of the MITgcm code since it continuously undergoes updates. It was configured to solve the Boussinesq, hydrostatic Navier–Stokes equations with a nonlinear free-surface (Marshall et al., 1997; Adcroft et al., 2004b). A cubed-sphere grid was used which provided a nearly uniform resolution and avoided pole singularities (Adcroft et al., 2004a). It consisted of 6 faces, each of which comprised 32 x 32 grid cells, resulting in a horizontal resolution of approximately 2.8°. There were 15 vertical levels, ranging in thickness

from 50 m at the surface to 690 m at the seafloor, giving a maximum model depth of 5200 m. Associated with the non-linear free-surface is the possible vanishing of the upper layer. To avoid this problem, the rescaled vertical coordinate z* was employed (Adcroft and Campin, 2004). This approach scales the entire vertical grid with the surface elevation and not just the

surface layer (cf. Fig. 1b in Adcroft and Campin, 2004). Furthermore, the shaved cell formulation was used, which reduced the representation error of the bathymetry (Adcroft et al., 1997). The model was coupled to a dynamic-thermodynamic sea ice model with a viscous-plastic rheology (Losch et al., 2010).

Isopycnal diffusion and eddy-induced mixing was parameterized with the GM/Redi scheme (Redi, 1982; Gent and McWilliams, 1990). Background vertical diffusivity for tracers was uniform at $3 \cdot 10^{-5}$ [$m^2 \, s^{-1}$], and for the equation of state the polynomial approximation of Jackett and McDougall (1995) was used. Advection of tracers was computed using third-order advection with direct space-time treatment (Hundsdorfer and Trompert, 1994).

Atmospheric forcing (air temperature, specific humidity, zonal and meridional wind velocity, wind speed, (snow-) precipitation, incoming shortwave and longwave radiation as well as river runoff – 12 climatological monthly means) was obtained from the PI ocean state estimate by Kurahashi-Nakamura et al., (2017), which was based on the protocol of the Coordinated Ocean-ice Reference Experiments (COREs) project (Griffies et al., 2009). They optimized the forcing fields to reconstruct tracer distributions that were consistent with observations. Air-sea fluxes were internally computed in the model following the bulk forcing approach by Large and Yeager (2004). Furthermore, we globally balanced the freshwater flux by annually adjusting the precipitation field (Appendix A).

Our simulation was initialized with present-day salinity and temperature distributions (Levitus et al., 1994 and Levitus and Boyer 1994, respectively) and spun up from the state of rest. We used asynchronous time stepping to accelerate computation with a time step of 1 day for the tracer equations and 20 min. for the momentum equations.

We compiled the code using the GNU Fortran compiler gfortran version 5.3.0 and performed the simulation on 6 cores of a processor of type Intel Xeon E5-2630 v3. The simulation was integrated for 3000 years (1000 model years took ~ 7.5 CPU hours) to reach a quasi-steady state (the global salinity, temperature and Atlantic Meridional Overturning Circulation were approximately steady at 34.73 psu, 2.86° C and 18.24 Sv (1 Sv = $10^6 \, m^3 \, s^{-1}$) respectively), continued for a further 3000 years with stable water isotopes as passive tracers. For analysis, the average of the last 100 years was used.

**2.2 Implementation of water isotopes**

We implemented the stable water isotopes $H_2^{16}O$, $H_2^{18}O$ and HDO as conservative, passive tracers in the ocean component of the MITgcm (wiso package). Isotopic variations at the sea surface were driven by evaporation ($E$), precipitation ($P$) and river runoff ($R$), while advection, diffusion and convection affected the distribution in the interior of the ocean. Monthly climatological means of the isotopic content of precipitation and water vapor were available from the National Center for Atmospheric Research Community Atmosphere Model including a water isotope scheme (NCAR IsoCAM – Tharammal et al., 2013). Note that the prescribed atmospheric forcing fields obtained from the PI ocean state estimate by Kurahashi-Nakamura et al., (2017) and the corresponding isotopic fluxes are not entirely consistent and might introduce an error in our model simulation. However, to minimize the uncertainty we only took the ratio of the isotopic content of precipitation and water vapor and applied it to the corresponding atmospheric forcing fields. The isotopic composition of river runoff affects the isotopic composition of ocean water ($\delta^{18}O_w$ and $\delta D_w$) particularly in coastal regions. Since there was no land model in the

MITgcm to calculate the amount and isotopic composition of continental runoff, we assumed that it equals the isotopic composition of the local precipitation at the river mouth and again applied it to the runoff forcing field.

Fractionation during evaporation, taking both equilibrium effects and kinetic effects into account, was treated explicitly in the MITgcm. The formulation for the isotopic composition of evaporation $E^i$ [mol m$^{-2}$ s$^{-1}$] is

$$E^i = \Gamma^i(q^i - q_s^i). \tag{2}$$

Here, q$^i$ is the specific humidity [kg kg$^{-1}$] multiplied by the isotopic ratio derived from the NCAR IsoCAM and

$$q_s^i = q_s \frac{J^i}{\alpha_{l-v}} \tag{3}$$

is the tracer specific humidity [kg kg$^{-1}$] in thermodynamic equilibrium with the liquid at the ocean surface (Merlivat and Jouzel, 1979), while

$$q_s = \frac{0.98}{\rho_{air}} q_{sat} \tag{4}$$

is the local sea surface humidity [kg kg$^{-1}$] with $q_{sat}$ being the saturation specific humidity [kg m$^{-3}$] and $\rho_{air}$ being the atmospheric density [kg m$^{-3}$],

$$J^i = \frac{c(i) \cdot M(i)}{c(H_2{}^{16}O) \cdot M(H_2{}^{16}O)} \tag{5}$$

is the local sea surface mass ratio with $c$ being the concentration [mol m$^{-3}$] and $M$ the molar mass [g mol$^{-1}$] of the respective stable water isotope. The equilibrium fractionation factor $\alpha_{l-v}$ between liquid water and water vapor has been found empirically as a function of temperature and was given by Majoube (1971):

$$\alpha_{l-v}^{\delta^{18}O} = e^{\frac{1.137}{SST^2} \cdot 10^3 - \frac{0.4156}{SST} - 2.0667 \cdot 10^{-3}} \tag{6}$$

$$\alpha_{l-v}^{\delta D} = e^{\frac{24.844}{SST^2} \cdot 10^3 - \frac{76.248}{SST} + 5.2612 \cdot 10^{-2}} \tag{7}$$

with SST being the sea surface temperature [K].

Due to different molecular diffusivities of the isotopes, kinetic fractionation occurs. The kinetic fractionation factor $K$ depends on wind speed $U$ [m s$^{-1}$] through the roughness of the air-sea interface (Merlivat and Jouzel 1979; Jouzel et al., 1987):

$$K_{H_2^{18}O} = \begin{cases} 0.006, & \text{if } U < 7 \, m \, s^{-1} \\ 0.000285 \cdot U + 0.00082, & \text{if } U \geq 7 \, m \, s^{-1} \end{cases} \tag{8}$$

$$K_{HDO} = \begin{cases} 0.00528, & \text{if } U < 7 \, m \, s^{-1} \\ 0.0002508 \cdot U + 0.0007216, & \text{if } U \geq 7 \, m \, s^{-1} \end{cases} \tag{9}$$

The kinetic fractionation factor was used to calculate the isotopic profile coefficient $\Gamma^i$ following:

$$\Gamma^i = \rho C_E U (1 - K) \tag{10}$$

where $\rho$ is the air density and $C_E$ is the transfer coefficient for evaporation as described in Large and Yeager (2004).

Fractionation during the formation of sea ice was neglected, because it is very small compared to other fractionation processes and thus only leads to minor effects on $\delta^{18}O_w$ and $\delta D_w$ (Craig and Gordon, 1965). Due to the absence of isotopes in the sea ice we approximated the isotopic surface flux $F^i$ [mol m$^{-3}$ s$^{-1}$] by:

$$F^i = -((E^i - P^i) \cdot (1 - A_{ice}) - R^i) \tag{11}$$

with $A_{ice}$ being the ice-covered area fraction. Based on this approximation, there was no isotopic surface flux in areas covered by sea ice unless they were influenced by river runoff. Within the MITgcm, processes that affected the stable water isotopes were taken care of by the "gchem" and "ptracers" packages (Table 1). While the "gchem" package acted as an interface between the ptracers and wiso package and added $F^i$ to the passive tracer surface tendency $gPtr^j$ [mol m$^{-3}$ s$^{-1}$]

$$gPtr^i = gPtr^i + F^i,$$ (12)

the "ptracers" package mainly accounted for the transport of the isotopes by advecting and diffusing them. Furthermore, due to the freshwater flux that effectively changed the water column height, an additional tracer flux $F_w^i$ [mol m$^{-2}$ s$^{-1}$] associated with this input/output of freshwater ($E - P - R$ [kg m$^{-2}$ s$^{-1}$]) was calculated following

$$F_w^i = (E - P - R) \cdot c_i \cdot x$$ (13)

with $x$ being an units conversion factor. $F_w^i$ was then additionally added to the tracer surface tendency within the "ptracers" package

$$gPtr^i = gPtr^i + F_w^i \cdot \frac{1}{z}$$ (14)

with $z$ [m] being the surface grid cell thickness.

In the MITgcm, the stable water isotopes were not treated as ratios, but as individual concentrations. Therefore, we initialized the ocean with homogenous concentrations of $H_2^{16}O$, $H_2^{18}O$ and $HDO$ matching present-day $\delta^{18}O_w$ and $\delta D_w$ values of 0‰ with reference to the VSMOW. The ratios were calculated during the analysis of the results.

Furthermore, similar to the freshwater flux, a correction factor for the tracer specific precipitation was applied, whereby the respective global tracer concentration in the ocean was conserved (cf. Appendix A).

## 2.3 Observational data

### 2.3.1 $\delta^{18}O_w$ data

The Goddard Institute for Space Studies (GISS) Global Seawater Oxygen-18 Database v1.21 comprises over 26,000 seawater $\delta^{18}O$ values collected since about 1950 (Schmidt et al., 1999) and therefore offers an opportunity to evaluate the modeled oceanic $\delta^{18}O$ values.

For comparison, we interpolated the GISS samples to the nearest tracer grid point of our model grid using inverse distance weighting. We excluded any data point with applied correction, from enclosed lagoons, representing estuarine or river data from near the coast or heavily influenced by meltwater, which means that we rejected all data points flagged as G, H, I, J, L and X in the GISS database (see Schmidt et al., 1999 for details – 23,232 data points remained). We could not expect our model to reproduce such conditions, based on our relatively coarse grid resolution.

Since the GISS data usually represent samples taken at a certain time during the year, we did not compare them to simulated annual mean isotope values. Instead, we used a long-term monthly mean value of the specific month, when the GISS sample was measured.

### 2.3.2 $\delta^{18}O_c$ data

Mulitza et al. (2003) compiled a number of $\delta^{18}O_c$ values measured on planktonic foraminifera from plankton tows (including data from Duplessy et al., 1981, Kahn and Williams 1981, Ganssen 1983, Bauch et al., 1997 and Peeters and Brummer, 2002). They limited their compilation on the four species *Globigerinoides ruber* white (*G. ruber* (w)), *Globigerina bulloides* (*G.*

*bulloides*), *Neogloboquadrina pachyderma* sinistral (*N. pachyderma* (s)) and *Globigerinoides sacculifer* (*G. sacculifer*), since these species are very abundant, cover a broad geographical and temporal range and belong to the shallowest-dwelling planktonic foraminifera. We extended this data set with available *in situ* $\delta^{18}O_c$ data from Kohfeld and Fairbanks (1996), Moos (2000), Stangeew (2001), Volkmann and Mensch (2001), Mortyn and Charles (2003), Keigwin et al. (2005), Wilke et al. (2009) and Rippert et al. (2016). By using inverse distance weighting, we interpolated the $\delta^{18}O_c$ data to the nearest tracer grid

point of the MITgcm grid (analogous to the GISS data) and compared them to the simulated long-term monthly mean $\delta^{18}O_c$ values of the respective month of sampling. We used the paleotemperature equation from Mulitza et al. (2004)

$$T\ [°C] = 14.32 - 4.28 \cdot (\delta^{18}O_c - \delta^{18}O_w) + 0.07 \cdot (\delta^{18}O_c - \delta^{18}O_w)^2 \tag{15}$$

to determine the dependency between the $\delta^{18}O_c$, the temperature $T$ during calcification and the $\delta^{18}O_w$. Since water samples are reported relative to the VSMOW standard and carbonate samples relative to the Vienna PeeDee Belemnite (VPDB) standard,

the $\delta^{18}O_w$ values need to be converted by subtracting -0.27 ‰ (Hut, 1987).

## 3 Results

### 3.1 General model performance – temperature and salinity distribution

We compare the simulated annual mean SST and sea surface salinity (SSS, upper 50 m) to the annual mean (averaged over the upper 50 m and interpolated to the cubed sphere grid) temperature (Fig. 1a, b) and salinity (Fig. 2a, b) of the World Ocean

Atlas 2013 (WOA13 – Locarnini et al. (2013), Zweng et al. (2013), respectively). In most regions of the World Ocean, SST differences are around 1 °C or even less (root mean square error (RMSE) = 1.18 °C) and therefore in good agreement with the data. Larger differences are mainly located in regions of coastal and equatorial upwelling, in the Gulf Stream and around Indonesia.

A different picture emerges for the SSS anomaly. While most parts of the surface ocean are slightly too fresh, especially the

Mediterranean Sea, Bay of Bengal, Hudson Bay and north of Iceland, both the Arctic Ocean and the east coast of North America are too salty. Nevertheless, we obtain a RMSE of 0.45 psu without using any salinity restoring.

This good agreement also continues in the deeper parts of the Atlantic Ocean. Calculated weighted zonal means of the simulated annual mean temperature and salinity in the Atlantic Ocean correspond well with the observations (Fig. 3a and b respectively; temperature and salinity provided by the GISS data – Schmidt et al., 1999). The simulated annual mean

temperature gradually decreases with depth, as do the observational data. It is also recognizable that the boundary towards water masses colder than 4 °C appears slightly shallower in the southern than in the northern part of the Atlantic Ocean. Coldest

temperatures occur in the deep southern Atlantic Ocean, both in the simulated as well as observational data. Interpolating the observational data to the nearest tracer grid point and comparing it to the simulated long-term monthly mean values of the respective month of sampling (as described in section 2.3.1 for the GISS data), further underlines the agreement between simulated and observed values (Fig. 3c – $r^2 = 0.93$, RMSE = 2.1 °C, n = 660). The zonally-averaged cross section of the simulated annual mean salinity clearly reveals the occurrence of different water masses. While most parts of the Atlantic Ocean are filled by the North Atlantic Deep Water (NADW) coming from the north with a salinity value of around 34.9 psu (reaching a water depth of ~ 3500 m), the deepest parts of the Atlantic Ocean basin are occupied by less saline water (~ 34.7 psu) of the Antarctic Bottom Water (AABW) flowing from the south. The Antarctic Intermediate Water (AAIW) is the freshest water mass (~ 34.6 psu) and can be traced as a tongue, spreading from the south towards the north at a water depth of 1000 m. The most saline water appears in the upper water column of the northern tropics (~ 30° N). This structure is also reflected in the observational data, however both NADW and AAIW seem to be slightly fresher. Performing a model-data comparison for salinity, as outlined above for temperature, shows a good fit (Fig. 3d – $r^2 = 0.61$, RMSE = 0.6 psu, n = 691) in general, but a few points are clearly located above the 1:1 line. These data points correspond to simulated annual mean salinity values in the upper water column near the North American coast, one of the regions with the highest positive SSS anomalies (Fig. 2a) and will be discussed shortly in section 4.1.

## 3.2 Stable water isotope distribution in ocean water

Even though measurements of δD exist, they are not as widespread as δ¹⁸O. Furthermore, the stable water isotope package will be used mainly for paleoclimatic reconstructions in conjunction with $\delta^{18}O_c$ data from benthic foraminiferal shells. Hence, we chose to focus on the comparison for δ¹⁸O to validate our simulation.

The surface (upper 50 m) distribution of annual $\delta^{18}O_w$ simulated by the MITgcm gradually decreases from the mid-latitudes to high latitudes (Fig. 4a, b). Highest values of about 1 ‰ occur in the subtropical gyre of the Atlantic Ocean, which are slightly higher than in the Pacific Ocean, reflecting the net freshwater transport by the trade winds. The Mediterranean Sea and Red Sea are regions of net evaporation and therefore contain $\delta^{18}O_w$ values of similar magnitude. The most depleted surface water is simulated in the high latitudes, showing values of -0.5 ‰ in the Southern Ocean and -1 ‰ in the Arctic Ocean. These depleted values result from negative $\delta^{18}O_w$ values in precipitation in combination with river/glacial runoff. Similarly, depleted values occur in surface waters around Indonesia.

The large-scale patterns and latitudinal gradients of simulated annual mean $\delta^{18}O_w$ values match fairly well the observations. For example, the model captures the contrast between high and low latitudes and the Atlantic and Pacific Ocean. However, some notable discrepancies are recognizable when comparing the absolute range of $\delta^{18}O_w$ at the surface. In the MITgcm, the subtropical gyres are less enriched than in the observations (annual mean value of 0.6 ‰ as compared to 1.0 ‰, respectively). The same holds true for the Mediterranean Sea. For the Arctic Ocean simulated $\delta^{18}O_w$ values are not as depleted as in the observational data (annual mean value -0.6 ‰ as compared to -1.5 ‰, respectively). Especially near large river estuaries, the model-data mismatch is large.

A clear distinction between different water masses based on the annual mean isotopic composition of sea water is recognizable in our simulation, both for the Atlantic Ocean and the Pacific Ocean (Fig. 5a and 5b respectively). In our model, the NADW in the Atlantic Ocean reaches down to approximately 3500 m depth and is rather enriched in $H_2^{18}O$, resulting in an annual mean $\delta^{18}O_w$ content of around 0.11 ‰ (cf. Table 3). Most enriched $\delta^{18}O_w$ values ($\sim$ 0.6 ‰) occur in the upper water column of the tropics (20° - 30° S and N). The NADW encounters the AAIW coming from the south at a water depth of approximately 1000 m. The latter is more depleted with an annual mean $\delta^{18}O_w$ value of around 0 ‰. The deepest parts of the Atlantic Ocean are characterized by negative annual mean $\delta^{18}O_w$ values of approximately -0.11 ‰ derived from AABW mixed with NADW. This water mass structure is in good agreement with the observational data. However, the NADW is not enriched enough compared to the observational data (0.21 ‰), whereby the deepest parts of the Atlantic Ocean reveal too depleted $\delta^{18}O_w$ values.

For the Pacific Ocean (Fig. 5b) the vertical structure is even more homogenous. Enriched waters ($\sim$0.1 ‰) occur in the upper water column down to approximately 1000 m. Deeper parts of the Pacific are filled with depleted water of around -0.1 ‰. Compared to the observational data, the vertical and latitudinal gradients are in agreement. The large number of negative $\delta^{18}O_w$ measurements at 50° N is obtained from the Okhotsk Sea and thus is not representative for a zonally-averaged cross section of the North Pacific.

To take a closer look at the model-data fit, we interpolated the GISS data to the nearest tracer grid point and compared it to the simulated long-term monthly mean value of the respective month of sampling (see section 2.3.1). The separation of the model-data comparison into different ocean basins (Atlantic Ocean - Fig. 6a, Pacific Ocean - Fig. 6b, Arctic Ocean - Fig. 6c and Indian Ocean - Fig. 6d) points to deviations that mainly occur in higher latitudes. The correlation and RMSE is quite diverse, showing strong correlation for the Indian ($r^2$ = 0.77, RMSE = 0.19 ‰, n = 593) and Pacific Ocean ($r^2$ = 0.74, RMSE = 0.32 ‰, n = 743), medium correlation for the Atlantic Ocean ($r^2$ = 0.37, RMSE = 0.79 ‰, n = 756) and no correlation for the Arctic Ocean ($r^2$ = 0.05, RMSE = 1.18 ‰, n = 1048). Overall, depleted $\delta^{18}O_w$ values are not very well simulated in the MITgcm, which is particularly recognizable in the Arctic, a region highly influenced by negative $\delta^{18}O_w$ values from precipitation, snow fall and river runoff (Yi et al., 2012).

**3.3 Relationship between stable water isotopes and salinity**

Similar physical processes determine the salinity and $\delta^{18}O_w$ distribution at the ocean surface. Thus, locally a linear relationship between those two quantities can be expected. Therefore, we compared the modeled $\delta^{18}O_w$-salinity relationship with the observed one by taking the closest long-term monthly mean tracer grid value of salinity and $\delta^{18}O_w$ to the GISS data points of the respective month of sampling. Restricting the comparison to the upper 50 m and the salinity range to 28 – 38 psu in order to reflect open ocean conditions, the general features of the latter relationship are well captured in our model (Fig. 7).

The modeled $\delta^{18}O_w$-salinity relationship in the tropics (25° S – 25° N) agrees quite well with the observed one (Fig. 7a). Here, we find a simulated slope of 0.15 ‰ psu$^{-1}$, while the observed one is 0.22 ‰ psu$^{-1}$. A steeper slope is visible in the mid-latitudes (25° S/N – 60° S/N) for both the simulated and observed relationship (Fig. 7b). However, the agreement between

those two slopes is smaller than in the tropics, with a simulated slope of 0.28 ‰ psu$^{-1}$ and an observed slope of 0.49 ‰ psu$^{-1}$. Further, it underlines that we do not simulate salinity and $\delta^{18}O_w$ values as low as represented in the GISS data.

### 3.4 $\delta^{18}O_c$ distribution

The annual mean simulated $\delta^{18}O_c$ distribution at the surface (upper 50 m) increases from the tropical regions (~ 3‰) to high latitudes (~ 3.5 ‰), reflecting the dependency on both $\delta^{18}O_w$ and temperature (Fig. 8). Most depleted $\delta^{18}O_c$ values develop in the Bay of Bengal and around Indonesia (< 3.5 ‰), while the transition towards positive $\delta^{18}O_c$ values occurs from 40° S/N upwards. Even though the plankton tow data are rather sparsely distributed in the global ocean, a latitudinal increase in $\delta^{18}O_c$ is also recognizable. Thus, the simulated large-scale pattern and latitudinal gradient match fairly well the measurements. Nevertheless, some model-data mismatch occurs. Simulated annual mean calcite values in the tropics seem to be slightly too low (e.g. northeast of the Amazon delta), while regions in the North Atlantic and Arctic Ocean are slightly enriched compared to the observations. The influence of the seasonal cycle on the $\delta^{18}O_c$ distribution depends on latitude (Fig. 9). The largest seasonal effects occur in the northern mid-latitudes (30° - 60° N) with values of up to 3 ‰, whereas a weak or almost no seasonal effect appears in low and high latitudes. Thus, when performing a model-data comparison, the respective month of plankton tow sampling must be considered. Figure 10a and b include not just the surface data but plankton tows taken in deeper parts of the ocean. The comparison reveals a good match (r² = 0.88, RMSE = 0.83 ‰, n = 183). Data points that are not located along the 1:1 line but rather above, belong either to the deeper water columns of the model (Fig. 10b) within the tropics (Fig. 10a) or, as mentioned above, to the upper water column (Fig. 10b) in high latitudes (Fig. 10a).

### 4 Discussion

### 4.1 Model performance

Before we discuss the $\delta^{18}O_w$ distribution in the MITgcm, the general model performance will be shortly assessed, because an accurate presentation of the ocean circulation is essential for a reasonable simulation of stable water isotopes. Therefore, we investigate the temperature and salinity distribution, because these two quantities determine the density and thus one of the main factors influencing the vertical movement of ocean waters. The results for the simulated annual mean temperature and salinity are quite promising. Large biases at the sea surface occur in the North Atlantic, both for the SST and SSS. These biases are quite common in ocean models, especially with a low resolution, since the proper simulation of the structure, pathways and extensions of western boundary currents are difficult to achieve (cf. Griffies et al., 2009). Here, the Gulf Stream remains attached to the coast far to north and due to the coarse grid resolution subpolar surface water displaces the North Atlantic Current resulting in SST and SSS biases. Regarding the SST, warm biases also occur in the upwelling regions along the west coasts of Africa and America (intruding far into the open ocean basin), which are mainly driven by the poorly resolved coastal upwelling process. In terms of SSS biases, surface boundary conditions like $P$ and $E$ should be considered. In general, the large-scale patterns for $P$ and $E$ are accurately presented (not shown here). The prescribed precipitation field $P$ clearly depicts

the intertropical convergence zones (ITCZ) in the Atlantic and Pacific Oceans. Further, extremely dry ocean regions in the subtropics that are associated with high pressure zones are visible. The simulated evaporation field $E$ is mainly zonally oriented, with increased values occurring in subtropical areas and decreased values along the equator and high latitudes. This zonal pattern is interrupted in regions of western boundary currents, where $E$ is enhanced along the pathways. For a more precise

estimate, we calculated annual anomalies for $P$ and $E$ (Fig. 11a and b, respectively) using data from rain gauge stations from the Global Precipitation Climatology Project (GPCP – Huffman et al., 1997) and the latent heat flux (converted to $E$ by dividing it with the constant latent heat of evaporation ($2.5 \cdot 10^6$ [J kg$^{-1}$] – Hartmann, 1994)) from the National Oceanography Centre (NOC) Version 2.0 Surface Flux and Meteorological Dataset (Berry et al., 2009). Unfortunately, no data exists for $E$ in high latitudes, whereby no model-data comparison can be carried out in these regions. Since $E$, among others, depends on the SST,

a similar picture for the anomaly should emerge. Indeed, regions with warmer (colder) SST simulated by the MITgcm also experience elevated (reduced) $E$ values. The precipitation however, is too small in the North Atlantic, the Bay of Bengal, the equatorial Atlantic and along 60° S, while too large mainly in the tropics (especially in the Pacific) and high latitudes. Regarding the SSS, the bias in the North Atlantic appears to be caused by an interaction between the coarse grid resolution and a bias in the evaporation. Besides the Mediterranean Sea, where enhanced $P$ and reduced $E$ lead to a fresh bias, there is no

other apparent correlation between $P$, $E$ and SSS anomalies. With a RMSE of 1.18 °C and 0.45 psu, respectively, our SST and SSS results are comparable to Danabasoglu et al. (2012), who reported a RMSE of 0.58 °C and 0.41 psu for the POP2 ocean component of the Community Climate System Model 4 (CCSM4) using a weak salinity restoring, and Griffies et al. (2011), who got a RMSE of 1.3 °C and 0.77 psu with the Geophysical Fluid Dynamics Laboratory Climate Model version 3.

Likewise, the comparison with observed data for the deep Atlantic Ocean basin is good. The main water masses AAIW,

NADW and AABW can be detected. Core properties of the water masses (AAIW: salinity = ~ 34.6 psu, temperature = ~ 5 °C; NADW: salinity = ~ 34.9 psu, temperature = ~ 3 °C; AABW: salinity = ~ 34.7 psu, temperature = ~ 0 °C; visual estimation based on Fig. 3) fit reasonably well the temperature and salinity ranges reported by Emery and Meincke (1986 – Fig. 14, rectangles). But both, NADW and AABW might be slightly too salty, while the AABW seems to be too cold. To maintain a realistic ocean climate, not just the water mass structure is of importance but also the circulation strength. The maximum

meridional transport at 48° N simulated in the MITgcm is 17.8 Sv, consistent with 16 ± 2 Sv reported by Ganachaud (2003) and Lumpkin et al. (2008).

Thus, we find that the general model performance is reasonable and comparable to both observations and other climate simulations.

## 4.2 Sources of error for δ¹⁸O$_w$

Results of the δ¹⁸O$_w$ distribution at the sea surface showed relatively large mismatches between modeled and observed data in the Arctic Ocean. As indicated by Eq. 11, there is no isotopic surface flux in areas that are covered by sea ice unless they are influenced by river runoff. Since parts of the Arctic Ocean are covered by sea ice all year round and others are seasonally influenced (not shown here), these areas do not experience any isotopic surface flux during most of the year. In this way, the

impact of precipitation and snow fall that is highly depleted is neglected, which could explain too high $\delta^{18}O_w$ values in the Arctic Ocean.

The spatial distribution of $\delta^{18}O_w$ in $P$ is also a matter of debate. The Global Network of Isotopes in Precipitation (GNIP – IAEA/WMO, 2010) provides a database with $\delta^{18}O_w$ in $P$ at more than 950 stations all around the globe. For the comparison with modeled annual $\delta^{18}O_w$ in $P$ only data with continuous sampling for a minimum of 5 years has been considered, resulting in 127 data points. Unfortunately, most of the data is continental, whereby a significant conclusion for the $\delta^{18}O_w$ in $P$ over the ocean is difficult. Nevertheless, all the main characteristics in $\delta^{18}O_w$ in $P$ can be identified (Fig. 12a). Due to the temperature effect on the fractionation during condensation (Dansgaard, 1964), $\delta^{18}O_w$ in $P$ decreases from mid- to high latitudes. While most enriched values occur in the regions of trade winds with slightly more depleted values along the ITCZ, the strongest depletion can be found over the polar ice sheets. For a more straightforward statement, we performed a model-data comparison (Fig. 12b) by interpolating the GNIP data to the closest tracer grid point of the MITgcm, revealing a good agreement between modeled and measured data ($r^2 = 0.72$, RMSE = 2.4 ‰, n = 91). Linking these results to the large $\delta^{18}O_w$ mismatches that emerged in the Arctic Ocean, subtropical gyres and the Mediterranean Sea let us conclude that the decreased $\delta^{18}O_w$ values at the ocean surface in the latter two regions are caused by an interaction of $P$, $E$ and $\delta^{18}O_w$ in $E$. Enhanced $P$ in the MITgcm has a dilutional effect on the water, while due to reduced $E$ not enough $^{16}O$ is removed from the ocean surface. $\delta^{18}O_w$ in $P$ seems to be reasonably well simulated. Unfortunately, we cannot compare our simulated $\delta^{18}O_w$ in $E$ to any observational data, but it could be that it is also slightly too enriched. Regarding the Arctic Ocean, except for the isotopic surface flux calculation as outlined above, insufficient river discharge and neglecting the fractionation during sea ice formation could be further sources for the model-data deviations. As part of the Pan-Arctic River Transport of Nutrients, Organic Matter and Suspended Sediments (PARTNERS) project, Cooper et al. (2008) published flow-weighted annual mean discharge and $\delta^{18}O_w$ data (collected between 2003 and 2006) from the six largest Arctic rivers (Table 2). According to their estimates, $\delta^{18}O_w$ values of Eurasian rivers decrease from west-to-east, thus the Ob' river discharges the most enriched freshwater (-14.9 ‰) while the water of the Kolyma river is most depleted in heavy isotopes (-22.2 ‰). This west-to-east trend is also recognizable in our model (Fig. 13b), where the Ob' river contributes freshwater with a $\delta^{18}O_w$ value of around -15.6 ‰ and the Kolyma river of around -20.5 ‰. Even though it seems that the isotopic composition of the Ob' river is too depleted, all the other three Russian rivers are not as depleted as seen in the PARTNERS data.

Measurements of the Yukon and Mackenzie rivers reveal intermediate isotopic signals (-20.2 ‰ and -19.1 ‰ respectively). In the MITgcm these signals are slightly more enriched with $\delta^{18}O_w$ values of around -17.1 ‰ and -18.9 ‰ for the Yukon and Mackenzie rivers respectively. A consideration of the overall river discharge to the Arctic Ocean reveals a slight underestimation as the flow-weighted average for all six rivers is -18.8 ‰, while in the model it is only -18.0 ‰. Not only does the isotopic signal of the river discharge matter, but also the discharge amount. Estimating the annual discharge amount in the MITgcm is difficult, because determining the grid cells that belong to the respective river is based on visually assigning them according to the location of the river mouth. This may lead to deviations compared to observational data. While simulated annual discharge for the Yenisey, Lena, Yukon and Mackenzie rivers is in good agreement with reported values by Cooper et

al. (2008 – Table 2), the amounts discharged by the Ob' and Kolyma rivers differ substantially. However, deviations of the annual discharge for all six rivers are tolerable (~ 400 km³ a$^{-1}$). Cooper et al. (2008) further reported that the Arctic Ocean basin receives 10 % of the global river runoff (1.3 Sv – Trenberth et al., 2007). The MITgcm fits right into this magnitude with 9.3 % of the simulated global river runoff (1.17 Sv) received by the Arctic Ocean (> 60° N). Thus, the deviations that appeared for the Ob' and Kolyma rivers are most likely attributable to the grid cell assignment described above and should not matter significantly. Therefore, both the isotopic signal of river runoff and the discharge amount are rather insignificant for the model-data mismatch in the Arctic Ocean. The general pattern of the simulated isotopic river discharge shows that river runoff is more enriched in low and mid-latitudes (Fig. 13a), which is in accordance with observations (IAEA, 2012). Thus, simulating the isotopic composition of river runoff by taking the isotopic composition of the local precipitation is a reasonable first approximation, but should be overcome by implementing a bucket model in the MITgcm, which calculates the river discharge and its isotopic content for individual catchment areas over land.

Further discrepancies between model and observations might be due to the formation and transport of sea ice. During the formation of sea ice, the heavier isotopes are entrapped in the solid ice structure, while depleted sea ice brine is expelled (O'Neil, 1968). However, this fractionation process is relatively small. Lehmann and Siegenthaler (1991) reported an equilibrium fractionation constant of 2.91 · 10$^{-3}$ between pure water and ice under equilibrium laboratory conditions, while Melling and Moore (1995) estimated a fractionation constant of 2.09 · 10$^{-3}$ for ~1 m thick ice in the Canadian Beaufort Sea. So even though sea ice is highly dynamic, excluding not only the fractionation during the formation of sea ice but also the transportation of isotopes within the sea ice might lead to minor local changes but should not be one of the main sources of error. Indeed, Brennan et al. (2013) investigated the impact of a fractionation factor for sea ice on $\delta^{18}O_w$ in the University of Victoria Earth System Climate Model (UVic ESCM). They conclude that local changes in $\delta^{18}O_w$ due to the contribution of sea ice are smaller than 0.14 ‰ and therefore rather negligible.

Furthermore, the coarse resolution of our model may cause some of the model-data discrepancies, since it is not able to resolve all of the physical processes. For instance, water that is transported towards the Nordic Seas as parts of the Gulf Stream System is displaced by water from the Labrador Sea due to the coarse horizontal grid system. Also, vertically the thermocline might not be as pronounced and located as in the real ocean, since e.g. the upper 500 m in the MITgcm are only represented by four layers. Observational data corresponding to depths within that transition layer might reflect a different signal than resolved by the ocean model.

Since $\delta^{18}O_w$ is a passive tracer, shifts at the ocean surface might propagate in the ocean interior. Errors in the general model performance might further add to the deviations in the deeper ocean. However, the water masses in the MITgcm in terms of structure, extent and magnitude are faithfully simulated (cf. 4.1) and thus can be ruled out as a significant error source.

Additionally, our isotopic forcing was not obtained from the same source as the atmospheric forcing for the freshwater, heat and momentum flux, whereby a maximum consistency cannot be ensured and an additional uncertainty to our sources of error is added.

Despite these sources of error, the simulated pattern of $\delta^{18}O_w$ both at the sea surface as well as in the deep ocean agrees fairly well with other recent studies such as the study by Xu et al. (2012) with an OGCM as well as the studies by Roche and Caley (2013) and Werner et al. (2016) with fully coupled models.

## 4.3 Water mass structure

The seawater oxygen isotope ratio and salinity are controlled by the same processes such as evaporation, precipitation, river runoff and sea ice formation. In this way, they are locally linearly related, resulting in a slope that varies between 0.1 ‰ psu$^{-1}$ in low latitudes and up to 1 ‰ psu$^{-1}$ in high latitudes. However, water that is evaporated from the ocean surface does not carry any salt, but stable water isotopes. The agreement between the simulated slope and observational slope in the tropical regions is good, but significantly weaker in the mid-latitudes. This mismatch is mainly caused by the stable water isotopes since the

overall comparison to observed SSS is quite good and comparable with other ocean models (cf. section 4.1).

Subtropical gyres are characterized by high salinity and $\delta^{18}O_w$ values. While the model shows reasonable salinities in these regions (Fig. 2a), its surface water is too depleted (Fig. 4a). As discussed in section 4.2 theses discrepancies rather stem from an interaction of reduced $E$ whereby not enough $^{16}O$ is removed from the ocean surface, $\delta^{18}O_w$ in $E$ that is probably slightly too enriched and the dilutional effect of enhanced $P$. As opposed to this are the values of low salinity and $\delta^{18}O_w$ at the other

end of the slope. They are mainly located around the upper boundary of the mid-latitudes (~ 60° N/S) near the coast (e.g. the Okhotsk Sea and Bering Sea) and within the western boundary currents (e.g. Gulf Stream). While the modeled salinity is slightly too salty, the $\delta^{18}O_w$ values are not as depleted as seen in observations, causing the deviations in the slope of the $\delta^{18}O_w$-salinity relationship. We infer, that the coarse grid resolution is the main driver for this mismatch.

Despite these discrepancies at the sea surface, the investigation of the water mass structure of the deeper parts of the ocean

reveals that the model is suitable to determine the large-scale distribution of water masses in terms of the $\delta^{18}O_w$ signature. Water mass formation regions are mainly located in the high-latitude Atlantic Ocean and produce large parts of the deep and bottom waters of the World Ocean. Hence, our investigation focuses on the main water masses (AAIW, AABW and NADW) within that basin. Emery and Meincke (1986) used published temperature and salinity data to determine the core properties of the main water masses of the World Ocean. Applying these characteristics of the Atlantic Ocean to both the GISS data and

modeled values (Fig. 14; Table 3), clearly shows the resemblance. All three water masses are found in the ocean model, but their temperature-salinity ranges differ slightly from those given by Emery and Meincke (1986) as discussed in section 4.1. Nevertheless, even though the absolute range of $\delta^{18}O_w$ values is narrower in the model than in the observations, the modeled mean values are remarkably close to the observations (cf. Table 3). Our results are quite encouraging, suggesting that the nonlinear free surface and real freshwater flux boundary conditions of the MITgcm indeed leads to an improvement compared

to other ocean models using salinity restoring (e.g. Paul et al., 1999; Xu et al., 2012).

Overall, even though some regions at the surface reveal localized biases regarding the $\delta^{18}O_w$ distribution, the water mass structure of the deeper parts of the ocean and their characteristic $\delta^{18}O_w$ values are successfully simulated. Hence, the ocean model is well suited to perform long-term simulations in a paleoclimatic context and investigating the respective $\delta^{18}O_w$ changes.

## 4.4 Planktonic foraminiferal $\delta^{18}O_c$

To address questions regarding the evolution and history of the ocean and climate, oxygen isotopic records derived from measurements of foraminiferal shells have been used extensively. Particularly, the last glacial maximum (LGM) and last deglaciation are time periods, for which the evidence comes from proxy data recorded as oxygen isotopes in $CaCO_3$. Hence,
before using the model for paleostudies, an evaluation of modeled and measured $\delta^{18}O_c$ for the PI climate is necessary.

The $\delta^{18}O_c$ of planktonic foraminifera is not only determined by $\delta^{18}O_w$ and temperature of the ambient water in which the calcification takes place, but also altered by vital effects and modifications after death. Vital effects involve, for example, the photosynthetic activity of algal symbionts. Species like *G. ruber* (w) and *G. sacculifer* harbour symbionts (Kucera, 2007) that change the microenvironment around the shell by increasing the calcification rates through $CO_2$ uptake and thus shifting the
pH towards more alkaline conditions corresponding to elevated carbonate ion concentrations ($[CO_3^{2-}]$). This mechanism will induce a kinetic fractionation that leads to relatively $^{18}$O-depleted shells (Ravelo and Hillaire-Marcel, 2007). Furthermore, in the course of ontogenesis successive shell chambers reveal more enriched $\delta^{18}O_c$ values (Bemis et al., 1998), while significant changes also occur during reproduction. Bé (1980), Duplessy et al. (1981) and Mulitza et al. (2004) as well as others argue that some planktonic foraminifera add an additional layer of calcite during reproduction (gametogenic calcification). This
additional calcite layer is secreted in deeper and cooler water masses, introducing an $^{18}$O enrichment in the shell. Duplessy et al. (1981) ascertained a $\delta^{18}$O mean enrichment of 0.78 ‰ and 0.92 ‰ in the shells of *G. ruber* and *G. sacculifer* from core-top sediments, respectively. Mulitza et al. (2004) also showed that foraminiferal shells from the sediment are increased in $\delta^{18}$O by approximately 0.5-1 ‰. The average $\delta^{18}$O composition recorded by a foraminiferal species at the sea floor is further influenced not only by the vertical migration within the water column, whereby signals from different depths are incorporated into the
foraminiferal shell, but also by seasonal variations in shell production. Species that prefer polar waters (e.g. *N. pachyderma* (s)) rather peak during summer, whereas species that are distributed in warm provinces (e.g. *G. bulloides*) reflect a spring signal followed by a smaller autumn peak (Kucera, 2007). Additionally, the isotopic composition of foraminiferal shells can also be altered after deposition due to dissolution. This is especially the case, if the initial shell is dissolved rather than the crust formed during gametogenesis (gametogenic calcite is often more resistant to dissolution (Bé et al., 1975)), further shifting
the $\delta^{18}$O towards higher values.

All these mechanisms described above cannot be captured in our model, because it does not have an ecosystem module included, which could represent the life cycle of foraminifera and factors that determine the incorporation of oxygen isotopes in foraminiferal shells. Neglecting these processes might lead to additional model-data discrepancies. To avoid them, a comparison with plankton tow data is more reliable for testing the general capability of the model to simulate $\delta^{18}O_c$, since the
depth and month of sampling is known (thus excluding any deviations due to seasonality or depth habitat) and the foraminifera are sampled alive (thus excluding any deviations due to gametogenic calcification or modifications after death).

For the surface distribution of $\delta^{18}O_c$, the largest discrepancies between model and data occurred in the Arctic Ocean. While the SST is too low, the $\delta^{18}O_w$ is not depleted enough in this region. These two effects could compensate each other, but the $\delta^{18}O_c$

reveals a slight overestimate, which results from the underestimated SST. To disentangle the background of any model-data mismatch it is best to investigate the model-data fit considering individual species (Fig. 15). Therefore, we use species-specific paleotemperature equations published by Mulitza et al. (2003 – Table 4). First, we notice that the correlation is weaker when individual species are considered compared to investigating them grouped together. The best model-data fit is captured for *G. bulloides* ($r^2$ = 0.72, RMSE = 0.65 ‰, n = 35), while it is significantly weaker for *N. pachyderma* ((s); $r^2$ = 0.41, RMSE = 0.71 ‰, n = 61). While the largest deviations for *N. pachyderma* (s) occur in the upper surface column, data points that deviate from the 1:1 line for the other three species mainly correspond to depths larger than 100 m (not shown here). This becomes clearer, when the model-data comparison is carried out for data that only falls in the upper level (< 50 m) of the ocean model, resulting in a significant improvement of the RMSE and $r^2$ for *G. ruber* ((w); $r^2$ = 0.86, RMSE = 0.41 ‰), *G. sacculifer* ($r^2$ = 0.80, RMSE = 0.37 ‰) and *G. bulloides* ($r^2$ = 0.83, RMSE = 0.56 ‰), while the RMSE worsens for *N. pachyderma* ((s); $r^2$ = 0.46, RMSE = 0.89 ‰). Even though the sampling depth of the plankton tow data is known and was used for interpolation to the respective grid cell, we suppose that the $\delta^{18}O_c$ signal recorded by the living foraminifera rather corresponds to a shallower water depth (at least for the first three species mentioned before). Schiebel and Hemleben (2005) illustrated the average depth inhabited by planktonic foraminifera (cf. Fig. 2 therein). While *G. ruber* (w), *G. sacculifer* and *G. bulloides* inhabit the upper surface column (~ 25 m, ~ 40 m and ~ 50 m, respectively), *N. pachyderma* (s) lives on average in deeper parts (~ 90 m) and thus might confirm the assumption above. Another source of error may be the coarse vertical resolution of the model.

Overall, modeled $\delta^{18}O_c$ values can be compared to data successfully with a better result when all species are grouped together compared to individual species. Taking into account the processes that potentially affect the $\delta^{18}O_c$ of foraminifera and considering the species-specific influence by habitat depth and seasonality, a comparison with $\delta^{18}O_c$ collected from sediment cores appears to be feasible in a future study.

## 5 Conclusions

Stable water isotopes have been successfully implemented in the MITgcm, using real freshwater and isotopic flux boundary conditions in conjunction with the non-linear free surface. The model captures well the broad pattern and magnitude of $\delta^{18}O$ in annual mean seawater, reflecting accurately regions of net evaporation. The most enriched surface water occurs in the subtropical gyre of the Atlantic Ocean, while the surface water in the Arctic Ocean is isotopically most depleted. However, the latter ocean basin is the one with largest model-data discrepancies. They mostly result from the absence of highly depleted precipitation and snow fall in areas covered by sea ice. The simulated $\delta^{18}O_w$-salinity relationship is in good agreement with observations in tropical regions but less so in mid-latitudes, due to the misrepresentation of $\delta^{18}O_w$ caused by the coarse grid resolution of the model as well as an interaction of *P*, *E* and $\delta^{18}O_w$ in *E*. But even though the $\delta^{18}O_w$ distribution at the sea surface reveals some deviations, the water mass structure of the deeper parts of the ocean and their characteristic $\delta^{18}O_w$ values are well captured in our model and show that $\delta^{18}O_w$ indeed can be used to characterize different water masses. Further, we tested simulated $\delta^{18}O_c$ against measurements of planktonic foraminiferal shells from plankton tow data. Again, the latitudinal

gradients and large-scale patterns are faithfully reproduced. The model-data fit is better when all species are grouped together, compared to individual species and the largest discrepancies are most likely attributable to different depth habitats. A better understanding of the factors that determine the recording of oxygen isotopes in foraminiferal shells might be provided by ecosystem models including foraminifera (Fraile et al. (2008); Lombard et al. (2009); Kretschmer et al. (2016)).

The MITgcm and its newly developed stable water isotope package offer a great opportunity to perform long-term simulations in a paleoclimatic context and assimilating water isotopes with the adjoint method. Thus, investigations of not only the respective changes in $\delta^{18}O_w$ but also in foraminiferal $\delta^{18}O_c$ during the LGM or last deglaciation can be performed.

## 6 Code availability

The water isotope package incorporated in the MITgcm can be obtained by contacting the first author: R. Völpel
(rvoelpel@marum.de). Additionally, a release of the package through the MITgcm repository will be prepared.

## Appendix A

The MITgcm provides a scheme that balances the freshwater flux (net fluxes are set to zero) at each time step, preventing uncontrolled drifts in salinity and sea surface height caused by an imbalance in precipitation, evaporation and runoff. However, this scheme adversely affects the seasonality of the net surface freshwater flux.

Following Large et al. (1997), a precipitation correction factor $f_P(y)$ (a tracer specific precipitation correction factor $f_p^i(y)$) is implemented in the MITgcm and computed each year $y$, whereby the global freshwater flux (the global isotopic flux) is annually balanced.

The correction factor is applied to the precipitation field (tracer specific precipitation field), such that the precipitation throughout a model year $y$ is given by:

$$P = f_P(y) \cdot P(y) \qquad\qquad\qquad\qquad A(1)$$

$$P^i = f_P^i(y) \cdot P^i(y) \qquad\qquad\qquad\qquad A(2)$$

The size of $f_P$ ($f_p^i(y)$) depends on the change in volume of global ocean freshwater throughout a year ($\Delta V^F_y$) (change in the amount of the global isotopic tracer in the ocean throughout a year - $\Delta n^i_y$) and the volume of precipitation falling on the ice-free ocean (amount of tracer specific precipitation) and river runoff (amount of tracer specific river runoff) as an annual integral
($V^P$ ($n^{P^i}$) and $V^R$ ($n^{R^i}$), respectively). These values are used to compute the correction factor for the following year:

$$f_P(y+1) = f_P(y)\left(1 - \frac{\Delta V^F_y}{(V^P + V^R)}\right) \qquad\qquad\qquad\qquad A(3)$$

$$f_P^i(y+1) = f_P^i(y)\left(1 - \frac{\Delta n^i_y}{(n^{P^i} + n^{P^i})}\right) \qquad\qquad\qquad\qquad A(4)$$

If the change in volume of global ocean freshwater is positive (negative), the global salinity will decrease (increase) and the correction factor is decreased (increased) for the next year (*y+1*). For the tracer specific correction factor, it applies that a positive (negative) change in the amount of the global isotopic tracer leads to an increase (decrease) in global tracer concentration and thus a decreased (increased) tracer specific correction factor for the next year (*y+1*). Throughout the model integration changes are getting smaller resulting in a precipitation correction factor (tracer specific precipitation correction factor) that remains approximately constant at $f_{P}(y) = 1.0014$ after ~ 1500 model years ($f_{P}^{H_2^{16}O}(y) = 1.0241$ after ~ 600 model years and $f_{P}^{H_2^{18}O}(y) = 1.0253$ after ~ 1200 model years - Fig. A1).

Acknowledgments.

We would like to thank T. Tharammal for providing the isotopic data of NCAR IsoCAM and T. Kurahashi-Nakamura for providing the atmospheric forcing fields obtained with the adjoint model. Further, we would like to thank Martin Losch for his advice throughout the model development. Comments and suggestions by the three anonymous reviewers and the Editor highly improved the quality and clarity of the manuscript. This project was funded through the DFG Research Center/Center of Excellence MARUM – "The Ocean in the Earth System".

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

**Table 1: Main packages involved in the simulation of the stable water isotopes and their respective purposes.**

| Package | Purpose |
|---------|---------|
| Ptracers | initializes, advects and diffuses the passive tracers |
| Gchem | interface between the ptracers and wiso package → takes care of the additional sources and sinks for the passive tracers (e.g. surface forcing) by calling the respective wiso routines and adding the isotopic surface flux $F^i$ to the tracer surface tendency $gPtr^i$ |
| Wiso | calculates the isotopic evaporation $E^i$ and surface flux $F^i$ |

**Table 2: Annual mean $\delta^{18}O$ of river runoff and discharge for each of the six largest Arctic rivers presented by Cooper et al. (2008) and simulated by the MITgcm. Note that the river runoff in the MITgcm is distributed along the coasts (Fig. 9a and b) and thus the distinction which grid cell belongs to which river is just a rough approximation and can cause discrepancies.**

| River | $\delta^{18}O$ [‰] simulated by the MITgcm | $\delta^{18}O$ [‰] by Cooper et al. (2008) | Annual Discharge [km³ a$^{-1}$] simulated by the MITgcm | Annual Discharge [km³ a$^{-1}$] by Cooper et al. (2008) |
|-------|-------|-------|-------|-------|
| Ob' | -15.6 | -14.9 | 779 | 373 |
| Yenisey | -17.7 | -18.4 | 475 | 656 |
| Lena | -19.8 | -20.5 | 508 | 566 |
| Kolyma | -20.5 | -22.2 | 457 | 114 |
| Yukon | -17.1 | -20.2 | 172 | 214 |
| Mackenzie | -18.9 | -19.2 | 276 | 322 |
| All Six Rivers | -18.0 | -18.8 | 2667 | 2245 |

**Table 3: $\delta^{18}O_w$ characteristics of the main water masses (Antarctic Intermediate Water – AAIW, North Atlantic Deep Water – NADW and Antarctic Bottom Water – AABW) in the Atlantic Ocean for the observational (GISS) and simulated data (MIT). The $\delta^{18}O_w$ characteristics are determined by applying the temperature and salinity ranges of the respective water masses, reported by Emery and Meincke (1986), to the data within in the Atlantic Ocean (basin mask is based on the WOA09).**

|  | AAIW | NADW | AABW |
|---|---|---|---|
| $\delta^{18}O_w^{GISS}$ range [‰] | -2.50 – 1.41 | -0.49 – 0.88 | -0.31 – 0.00 |
| $\delta^{18}O_w^{GISS}$ mean value [‰] | -0.09 | 0.21 | -0.14 |
| $\delta^{18}O_w^{GISS}$ standard deviation [‰] | 0.42 | 0.09 | 0.08 |
| $\delta^{18}O_w^{MIT}$ range [‰] | -0.25 – 0.10 | 0.02 – 0.14 | -0.16 – -0.03 |
| $\delta^{18}O_w^{MIT}$ mean value [‰] | 0.00 | 0.11 | -0.11 |
| $\delta^{18}O_w^{MIT}$ standard deviation [‰] | 0.07 | 0.03 | 0.06 |

**Table 4: Data-model comparison of $\delta^{18}O_c$ of planktonic foraminifera data using species specific palaeotemperature equations (Mulitza et al. (2003)).**

| Foraminiferal species | Palaeotemperature equation | RMSE [‰] | $r^2$ | slope [‰ ‰$^{-1}$] |
|---|---|---|---|---|
| *G. ruber* (w) | $T = -4.44 \cdot (\delta^{18}O_c - \delta^{18}O_w) + 14.20$ | 0.89 | 0.41 | 0.77 |
| *G. sacculifer* | $T = -4.35 \cdot (\delta^{18}O_c - \delta^{18}O_w) + 14.91$ | 0.81 | 0.44 | 0.97 |
| *G. bulloides* | $T = -4.70 \cdot (\delta^{18}O_c - \delta^{18}O_w) + 14.62$ | 0.65 | 0.71 | 1.05 |
| *G. pachyderma* (s) | $T = -3.55 \cdot (\delta^{18}O_c - \delta^{18}O_w) + 12.69$ | 0.71 | 0.41 | 0.53 |

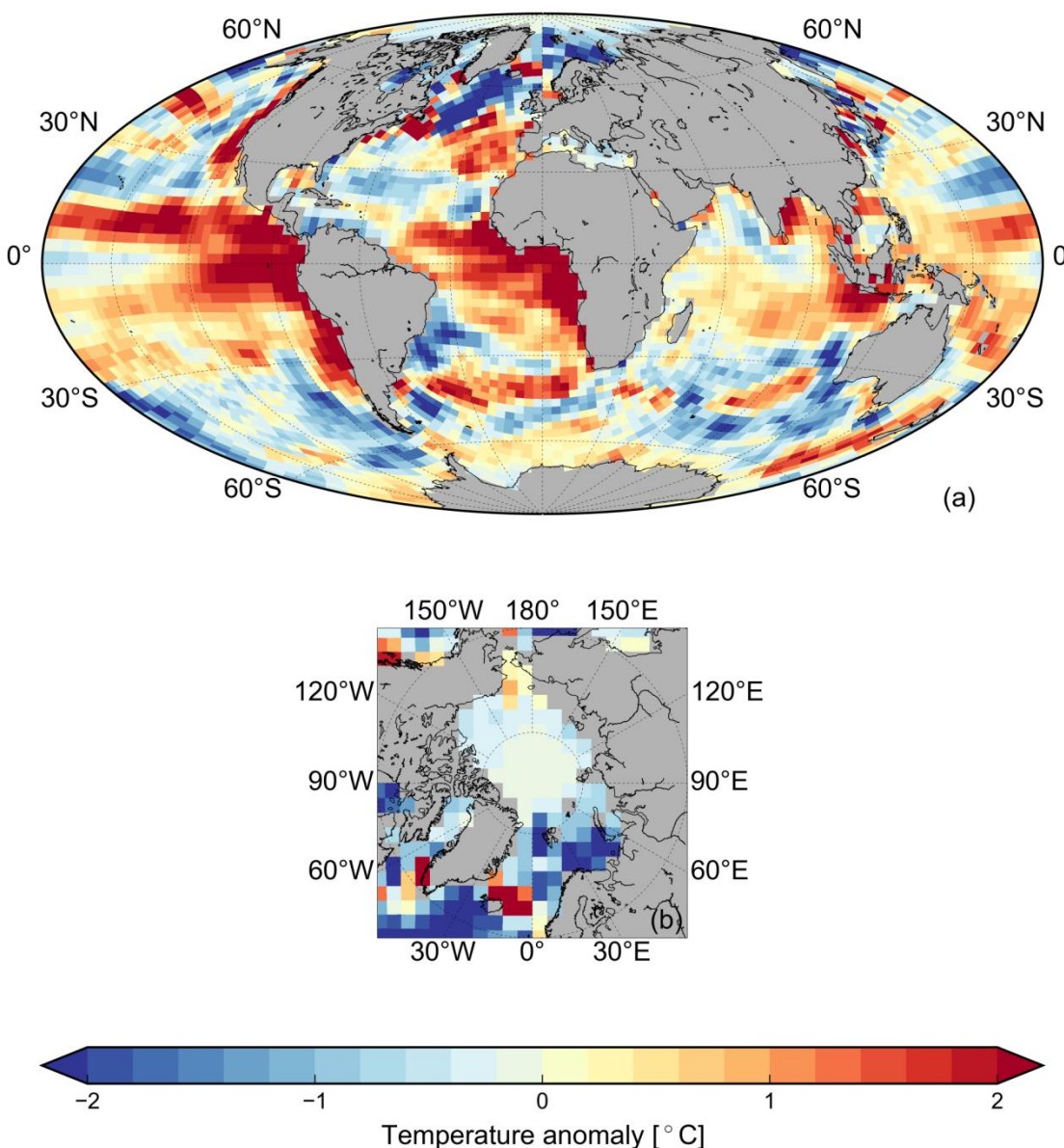

**Figure 1: Annual mean sea surface temperature anomaly (MITgcm – WOA13, upper 50 m) for (a) the global ocean and (b) the Arctic Ocean. For the calculation of the anomaly the SST of the WOA13 was averaged over the upper 50 m and interpolated to the cubed sphere grid of the MITgcm.**

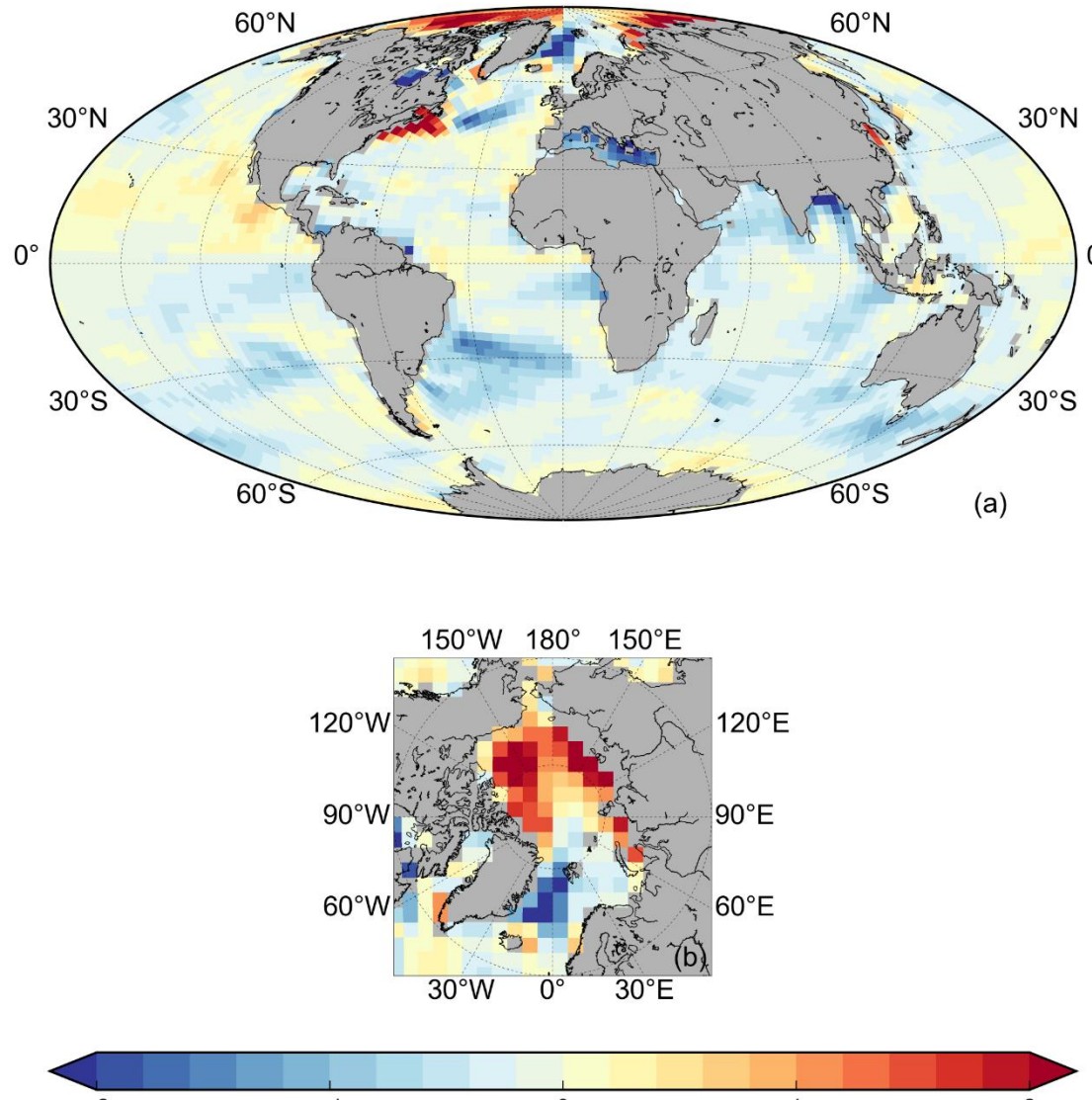

**Figure 2: Annual mean sea surface salinity anomaly (MITgcm – WOA13, upper 50 m) for (a) the global ocean and (b) the Arctic Ocean. For the calculation of the anomaly the SSS of the WOA13 was averaged over the upper 50 m and interpolated to the cubed sphere grid of the MITgcm.**

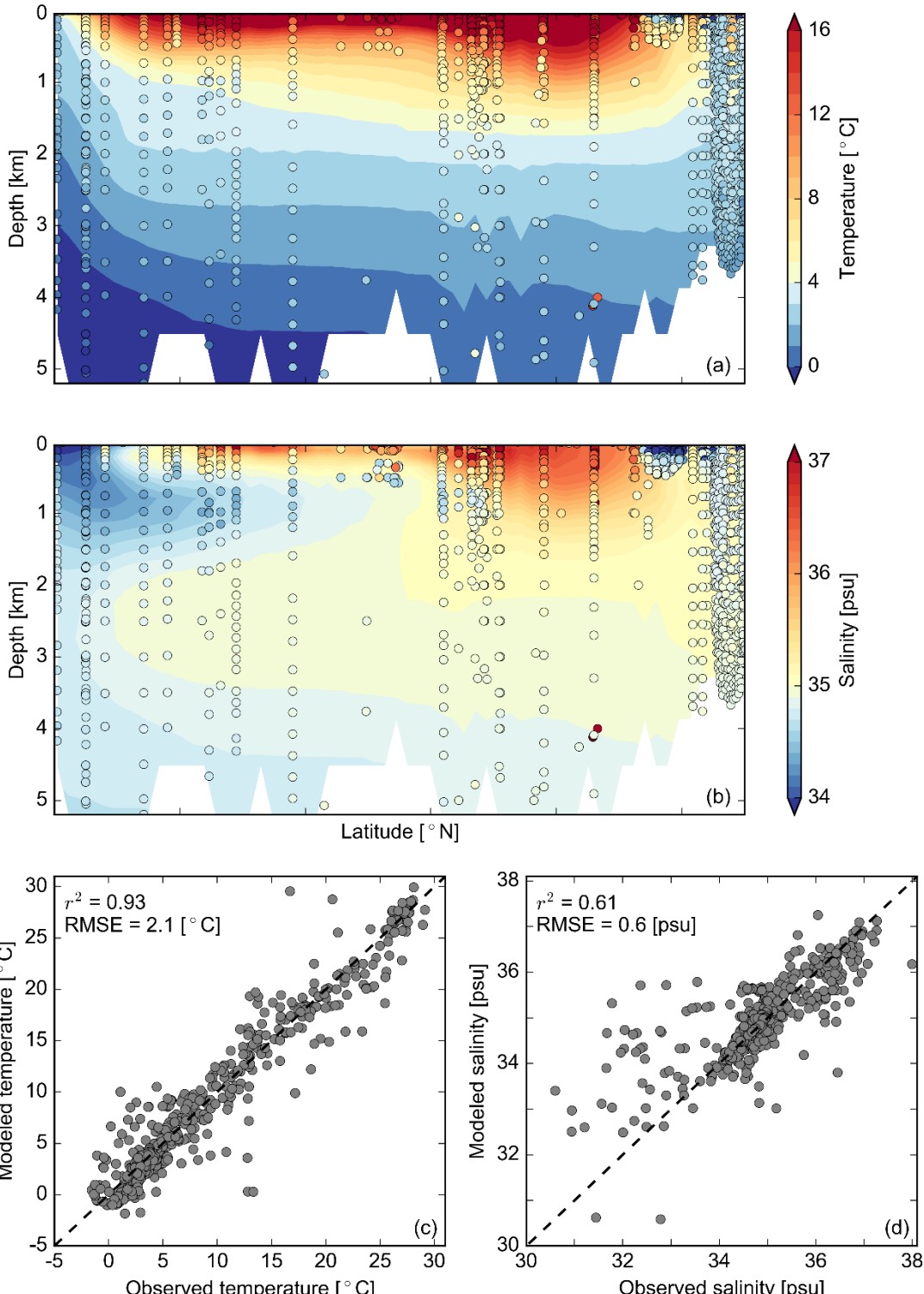

**Figure 3: Zonally-averaged cross sections through the Atlantic Ocean for (a) the simulated annual mean temperature distribution and (b) the simulated annual mean salinity distribution in comparison to the observational GISS data (colored symbols – Schmidt et al., 1999; (a): n = 2234, (b): n = 2666). The zonal-averaged cross sections have been determined using the Atlantic basin mask provided by the WOA09 (Locarnini et al., 2010) and dividing it into equally spaced latitudinal bands along which a weighted zonal mean was calculated. Note that the GISS data does not represent a zonal mean, but rather values from specific locations taken at a certain time during the year. The relationship between the observed data and simulated long-term monthly mean temperature and salinity in the Atlantic Ocean is presented in (c) and (d) respectively. For the comparison, the specific month of GISS sampling has been considered. Dashed lines represent the 1:1 line.**

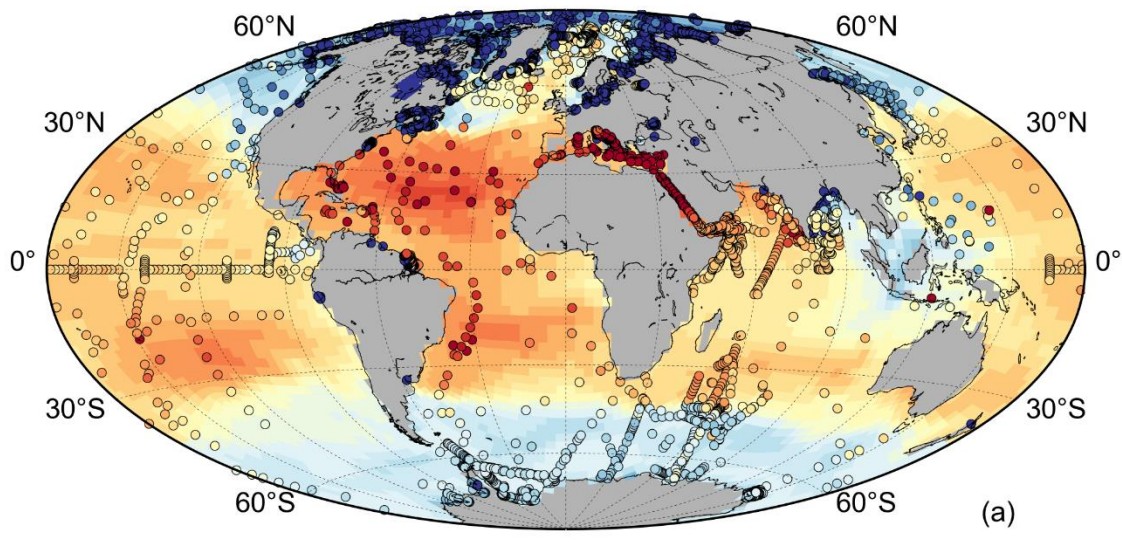

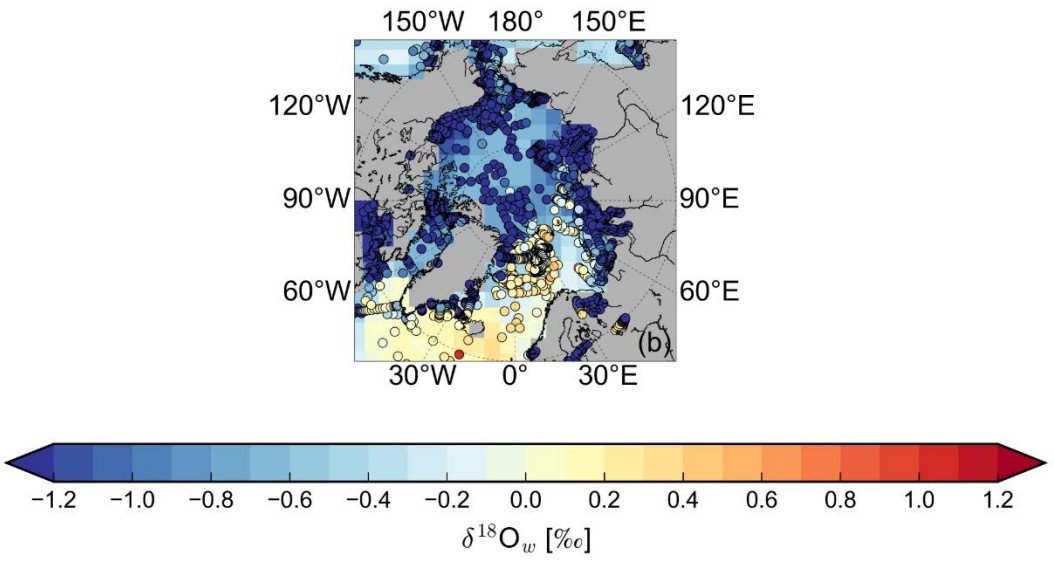

**Figure 4: Global annual mean surface (upper 50 m) δ¹⁸Oᵥ distribution simulated by the MITgcm in comparison to the observational GISS data (colored symbols - Schmidt et al., 1999) for (a) the global ocean and (b) the Arctic Ocean. The GISS data are averaged over the upper 50 m and do not represent an annual mean, but a certain time during the year.**

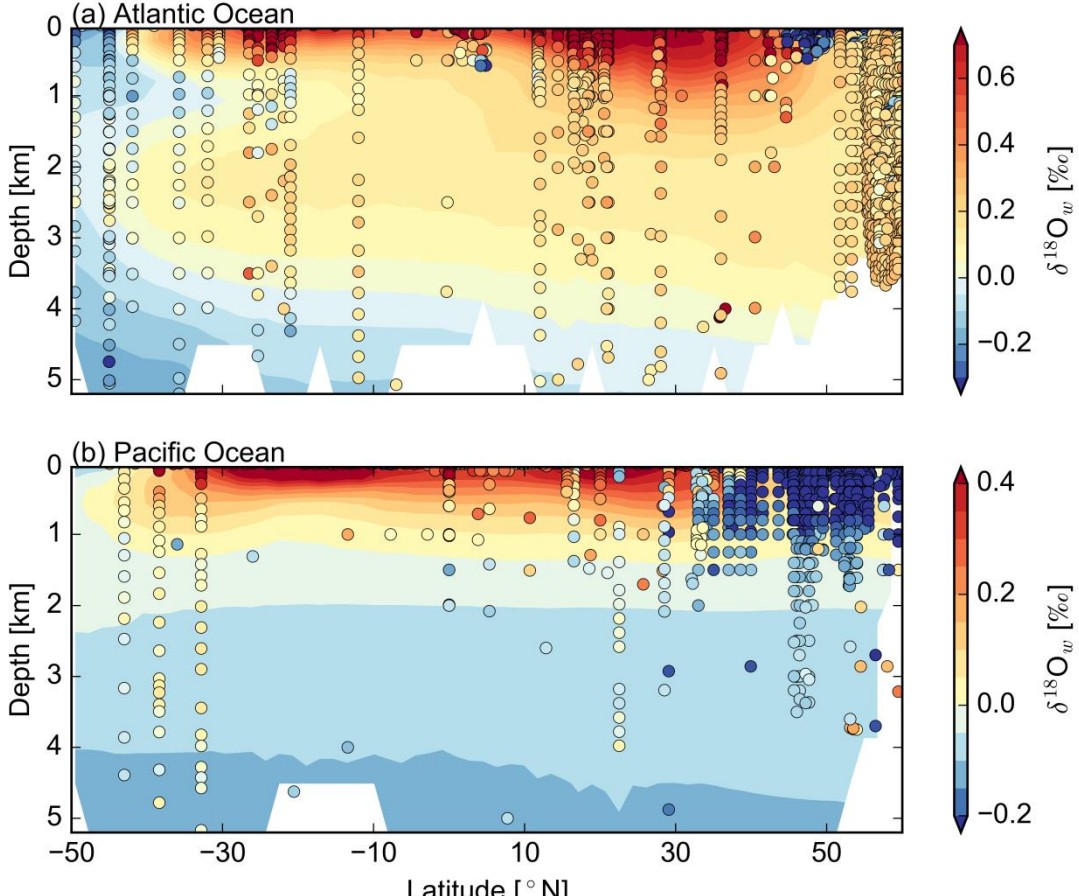

**Figure 5: Zonally-averaged cross section for the simulated annual mean δ¹⁸Oᵥᵥ distribution in (a) the Atlantic and (b) the Pacific Ocean in comparison to the observational GISS data (colored symbols – Schmidt et al., 1999; Atlantic Ocean: n = 2713, Pacific Ocean: n = 2929). The zonal-averaged cross sections have been determined using the respective basin masks provided by the WOA09 (Locarnini et al., 2010) and dividing it into equally spaced latitudinal bands along which a weighted zonal mean was calculated. Note that the GISS data does not represent a zonal mean, but rather values from specific locations taken at a certain time during the year.**

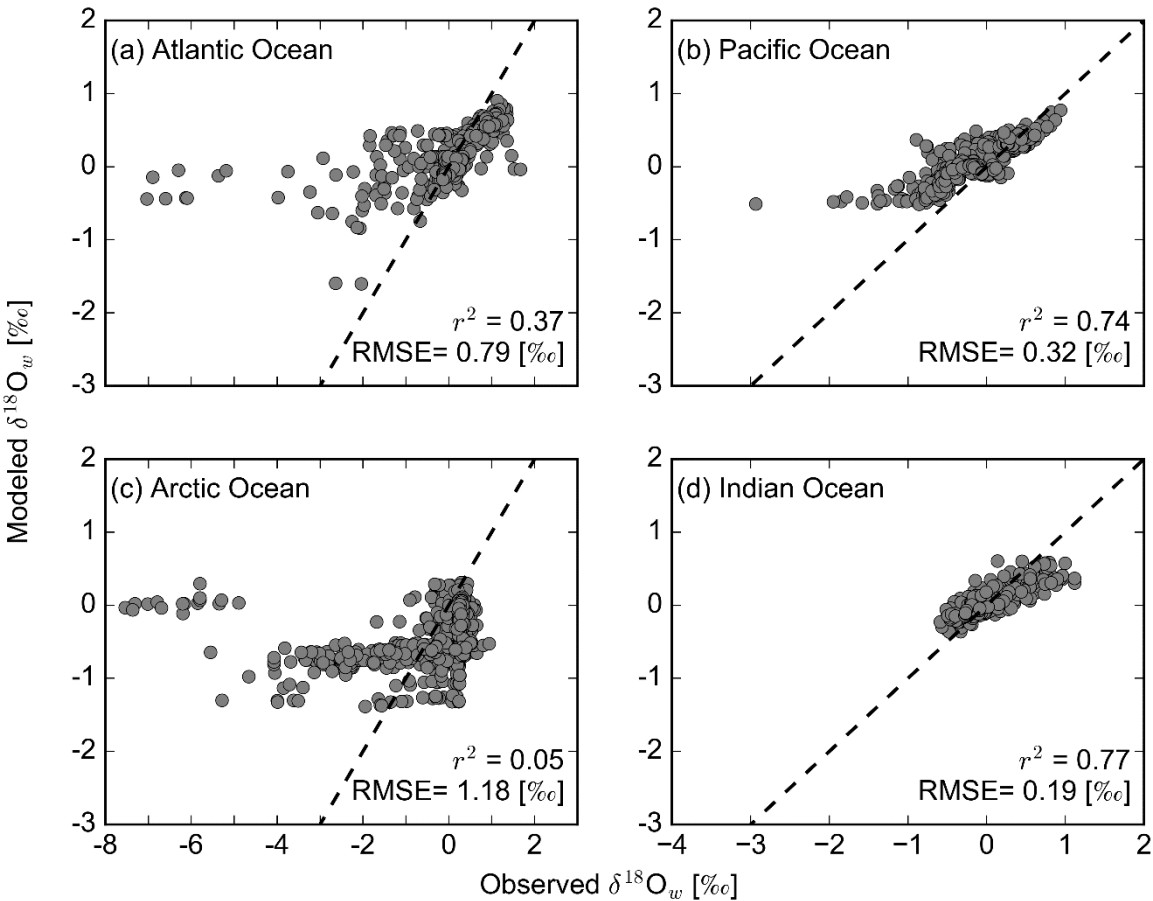

**Figure 6: Relationship between observed δ¹⁸Oᵥᵥ from the GISS database (Schmidt et al., 1999) and simulated long-term monthly mean δ¹⁸Oᵥᵥ from the MITgcm for the different ocean basins: (a) Atlantic Ocean, (b) Pacific Ocean, (c) Arctic Ocean and (d) Indian Ocean. For the comparison, the specific month of GISS sampling has been considered. Dashed lines represent the 1:1 line.**

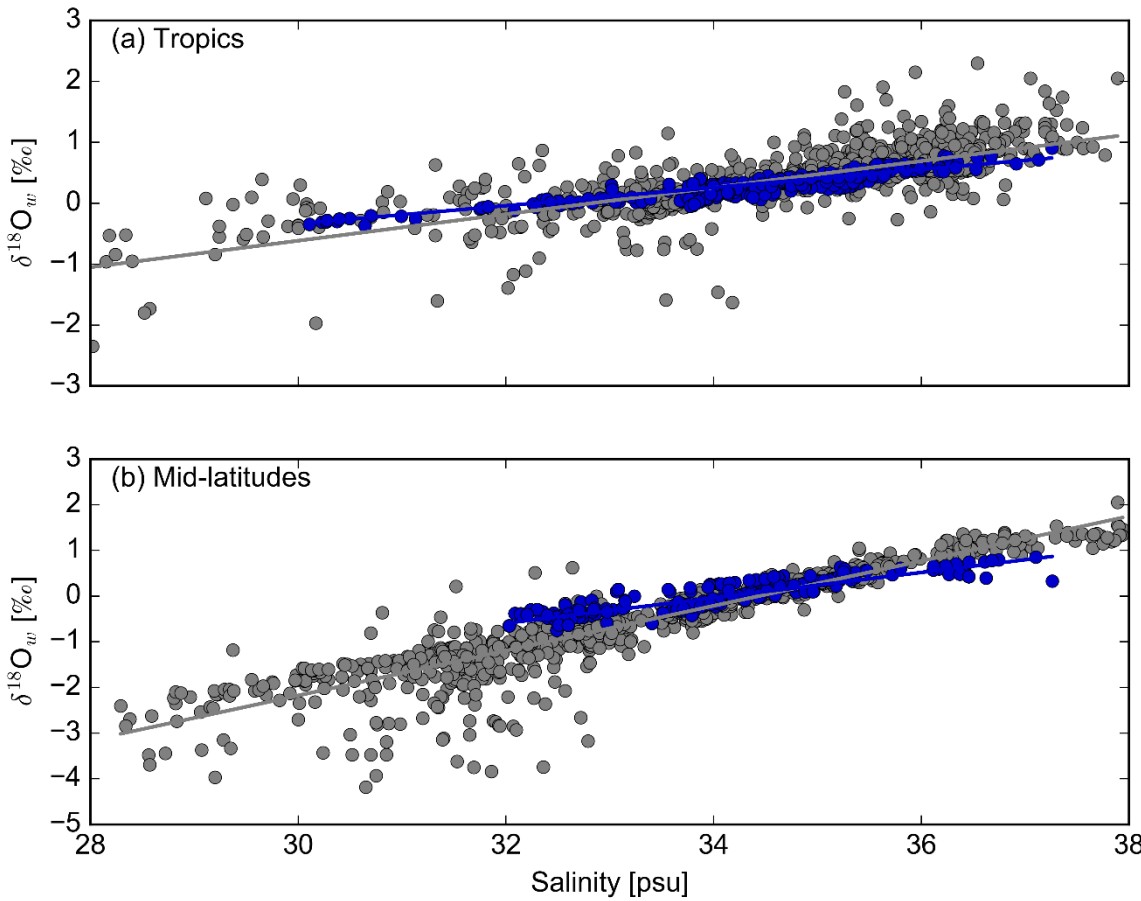

**Figure 7: Salinity and δ$^{18}$O$_w$ relation in surface waters (upper 50 m) for observational data (grey symbols – Schmidt et al., 1999) and simulated values (blue symbols) in (a) the tropics (25°S – 25°N) and (b) the mid-latitudes (25°S/N – 60°S/N). All GISS data in a depth range of 0-50 m with both salinity and δ$^{18}$O$_w$ values available are presented (tropics: n = 1191, mid-latitudes: n = 1282), while the closest long-term monthly mean tracer grid value of salinity and δ$^{18}$O$_w$ to the GISS datapoints of the respective month of sampling were chosen (tropics: n = 292, mid-latitudes: n = 245). The δ$^{18}$O$_w$/salinity slopes are given in the text.**

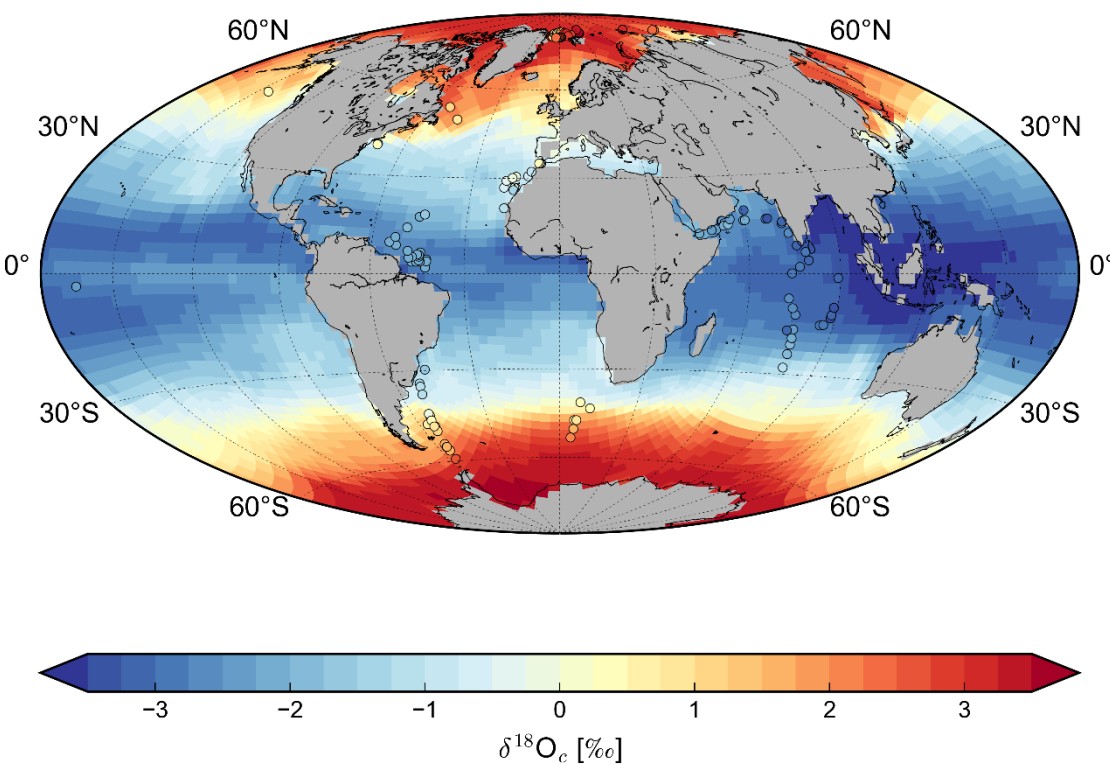

**Figure 8: Modeled annual mean sea surface δ¹⁸O_c distribution (upper 50 m) compared to δ¹⁸O_c values measured on planktonic foraminifera from plankton tows (colored symbols – for references see text).. The plankton tow data are averaged over the upper 50 m and do not represent an annual mean, but a certain time during the year.**

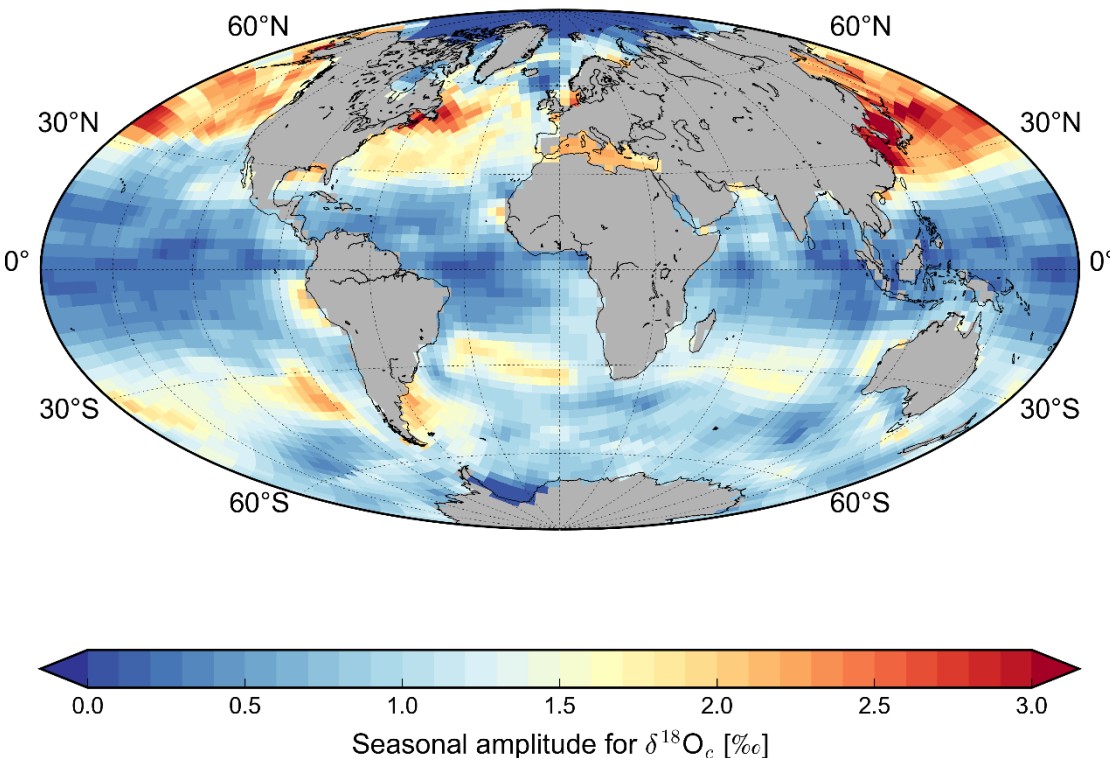

**Figure 9: Simulated seasonal amplitude for δ¹⁸Oc at the surface (upper 50 m). The seasonal amplitude is determined by calculating the absolute value between the two extreme months.**

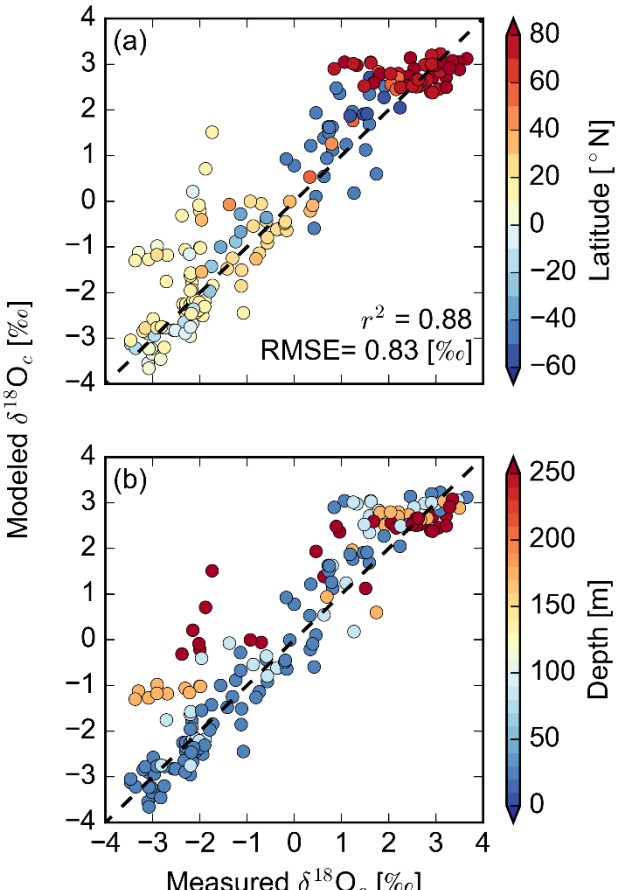

**Figure 10: Relationship between measured δ¹⁸O_c from various planktonic foraminiferas from plankton tows (for references see text) and simulated long-term monthly mean δ¹⁸O_c from the MITgcm either depending on latitude (a) or depth (b). For the comparison, the specific month and depth of plankton tow sampling has been considered and plankton tow data was interpolated to the closest tracer grid cell of the model, using inverse distance weighting. Dashed lines represent the 1:1 line.**

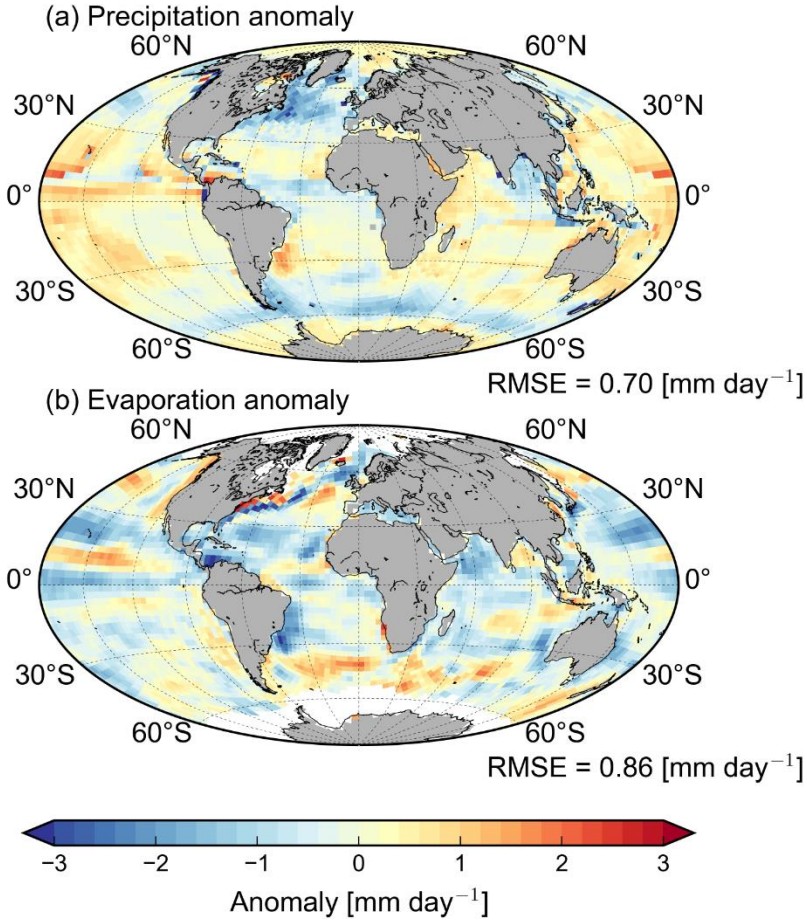

**Figure 11: Annual mean precipitation (a) and evaporation (b) anomaly (MITgcm – observational data). The observed precipitation field is provided by GPCP (Huffmann et al., 1997), while the latent heat flux from the NOC Version 2.0 Surface Flux and Meteorological Dataset (Berry et al., 2009) is converted to evaporation and used for comparison.**

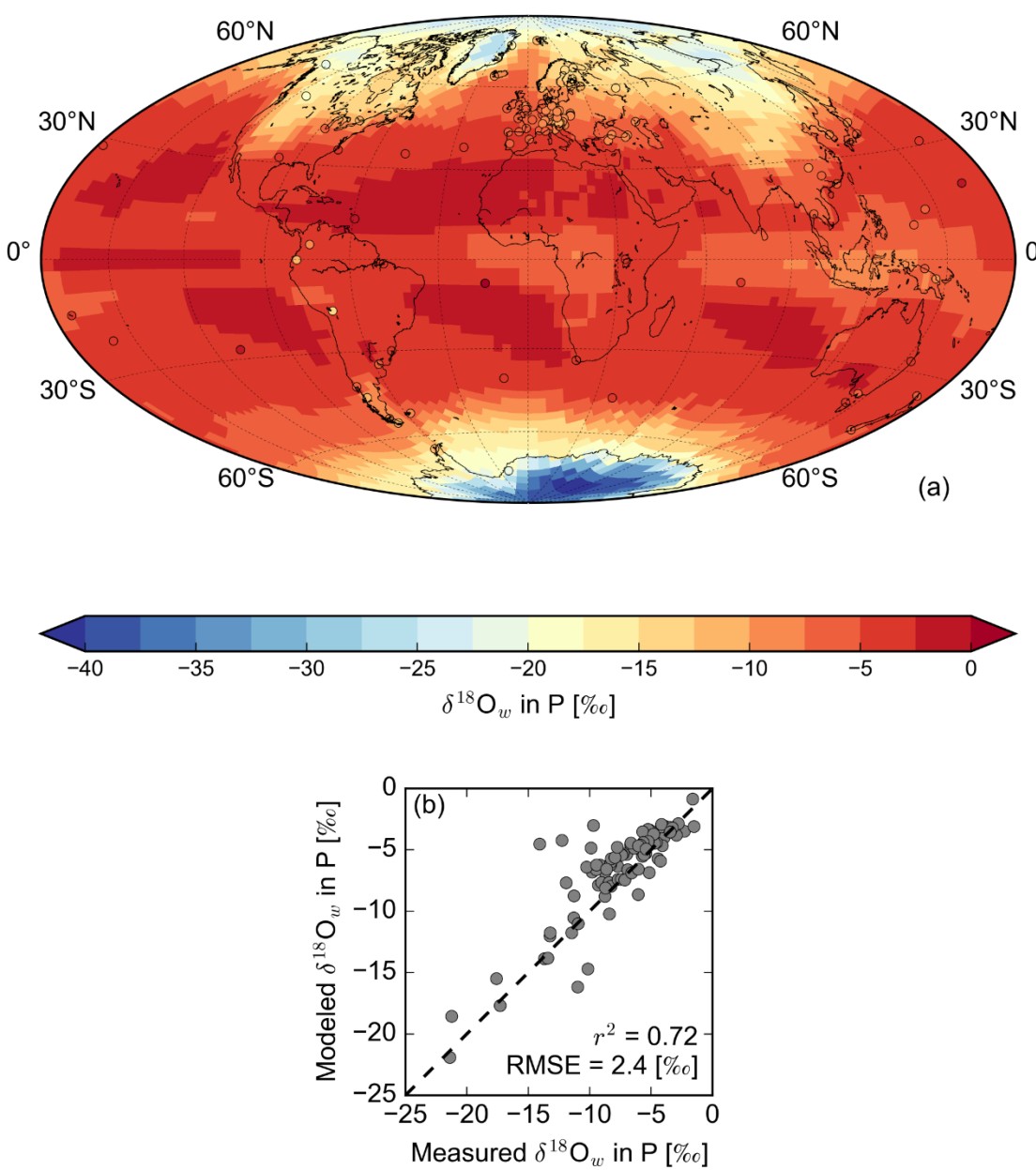

**Figure 12: (a)** Prescribed annual mean isotopic composition in precipitation compared to GNIP data (colored symbols IAEA/WMO, 2010). **(b)** Model-data comparison of the annual mean values. GNIP data was interpolated to the closest tracer grid cell of the MITgcm, using inverse distance weighting. Dashed lines represent the 1:1 line.

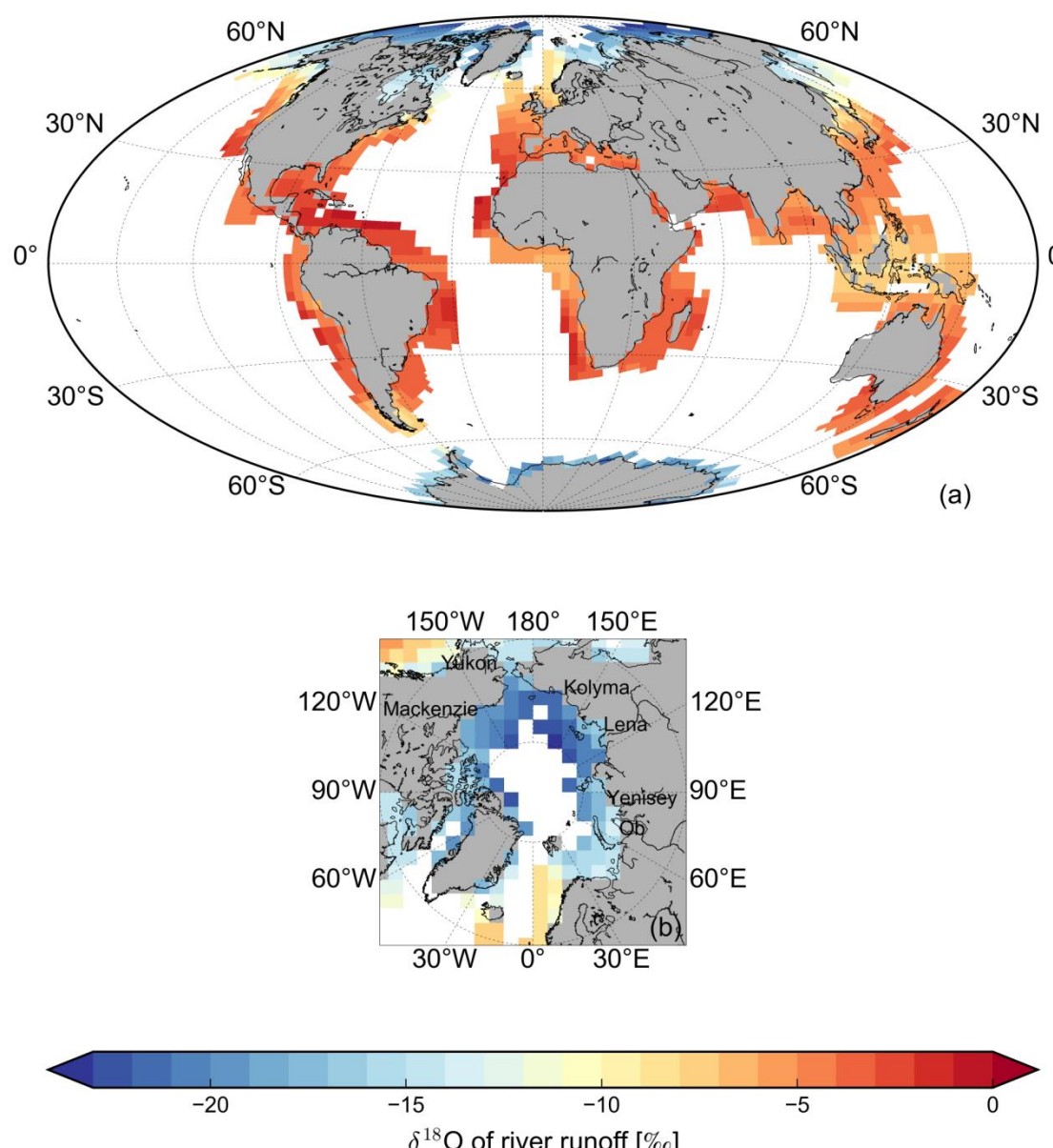

**Figure 13: Simulated annual mean δ¹⁸O of river runoff in the upper 50 m for (a) the global ocean and (b) the Arctic Ocean with the approximate location of discharge of the six largest rivers.**

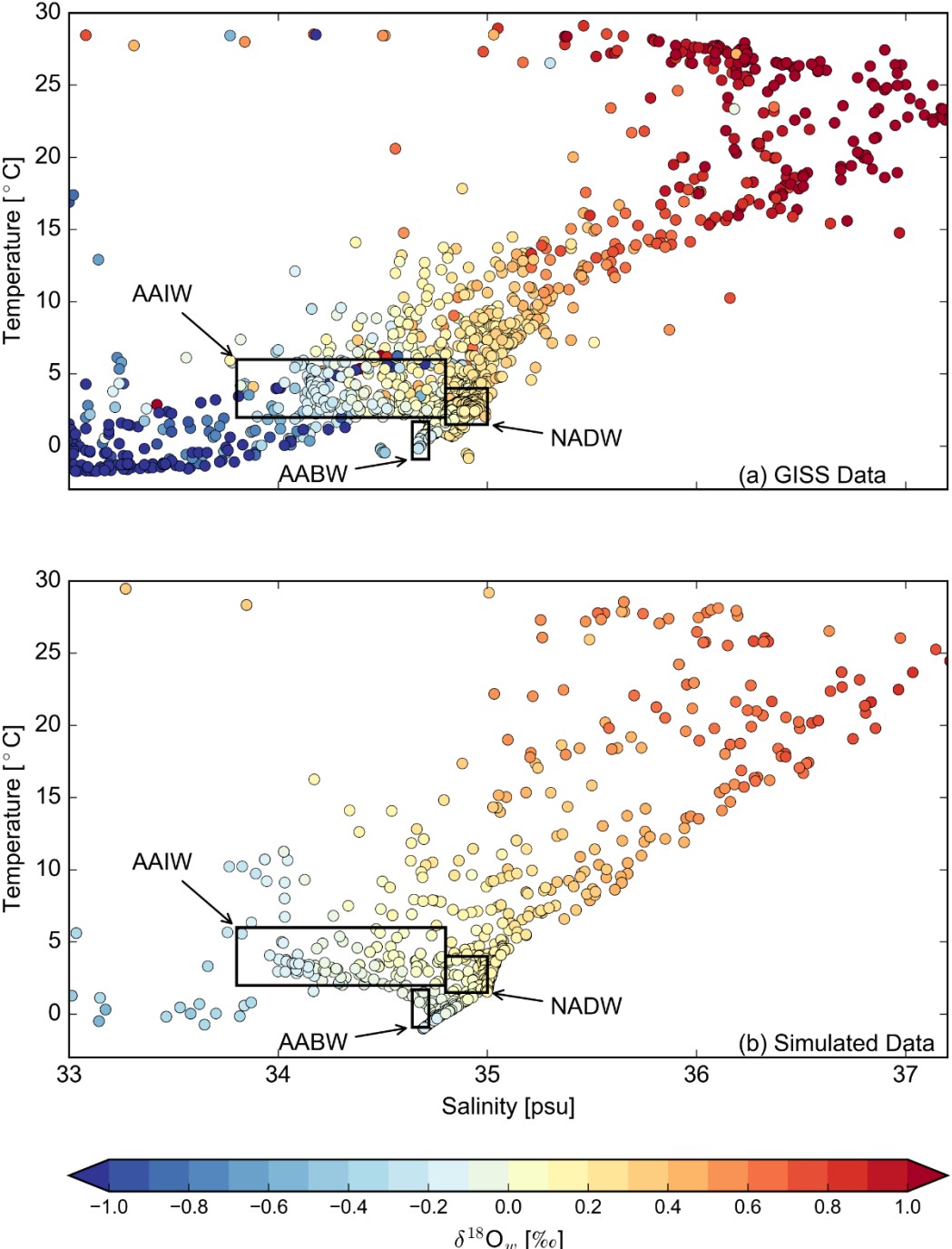

**Figure 14: Combined T-S-δ¹⁸Oₓ diagrams for the (a) GISS data and (b) simulated data (annual mean) in the Atlantic Ocean. The temperature and salinity ranges for the different water masses in the Atlantic Ocean are defined according to Emery and Meincke (1986).**

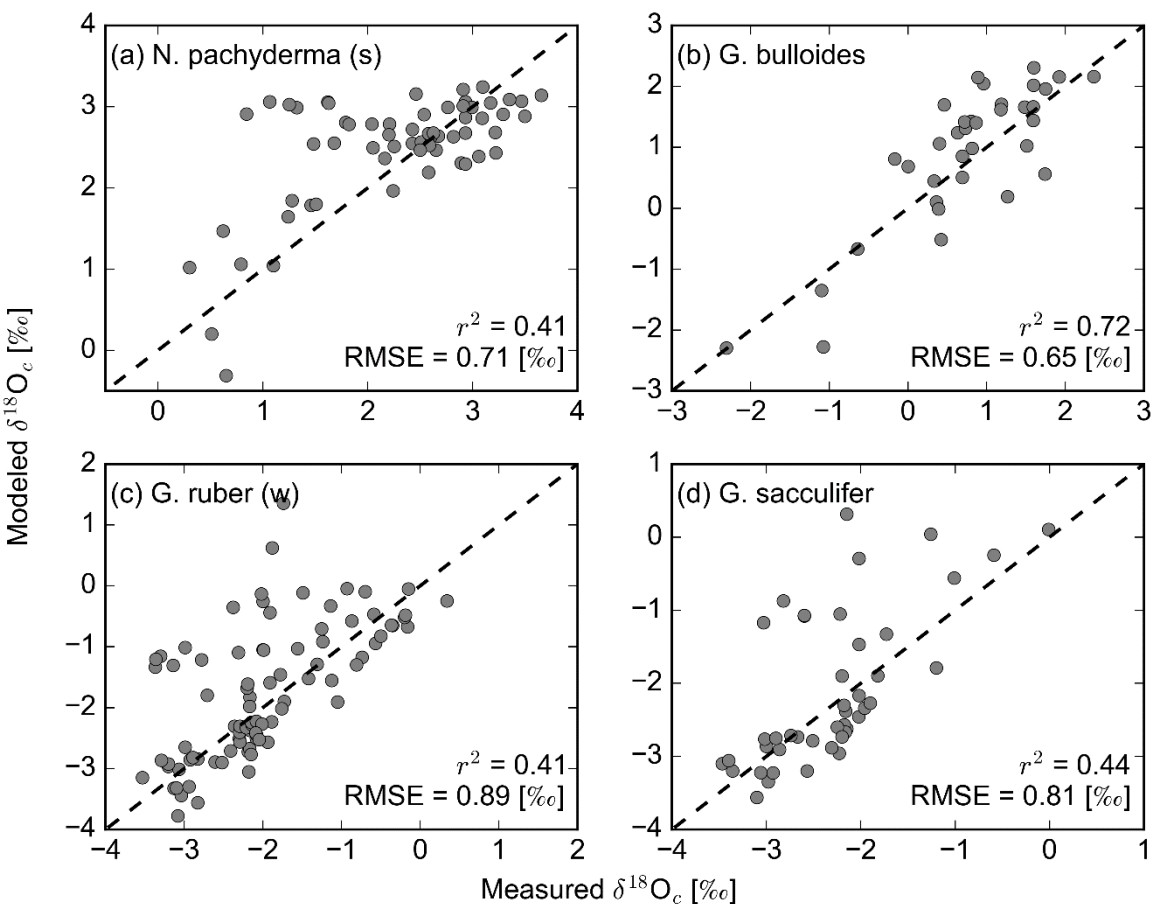

**Figure 15: Relationship between measured $\delta^{18}O_c$ from plankton tow data (for references see text) and simulated long-term monthly mean $\delta^{18}O_c$ from the MITgcm for the individual species: (a) *N. pachyderma* (s), (b) *G. bulloides*, (c) *G. ruber* (w) and (d) *G. sacculifer*. For the comparison the specific month and depth of plankton tow sampling has been considered. Dashed lines represent the 1:1 line.**

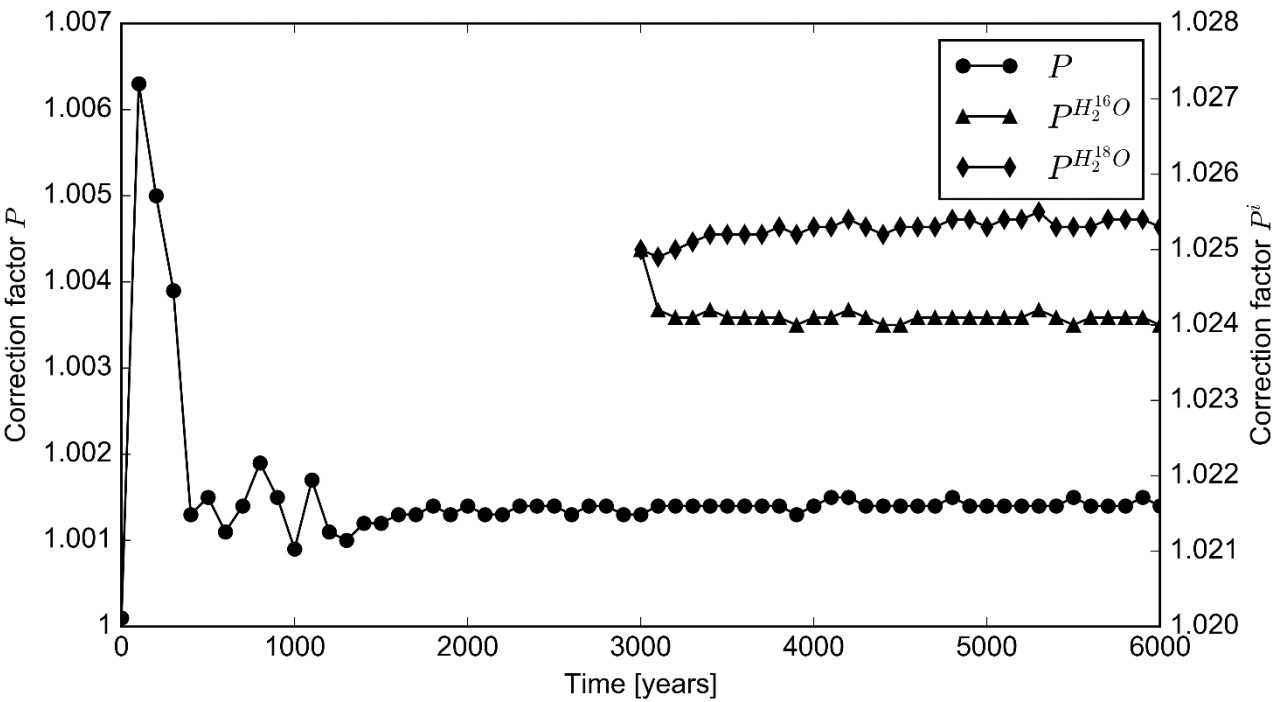

**Figure A1: Time series of the correction factor for both, the precipitation and tracer specific precipitation throughout the model integration.**