# Peer review of "Stable water isotopes in the MITgcm"

_Geoscientific Model Development, 2017_

## Referee Comment (RC1) · Anonymous Referee #1 · 9 Mar 2017

General comments Völpel et al., present the first results of the implementation of oxygen stable isotopes in the ocean general circulation model MITgcm. They compare the results of an equilibrium simulation under pre-industrial conditions to observational oxygen stable isotopes data and late Holocene data from planktonic foraminifera. They discuss the accordance or discrepancy of this data-model comparison. I find always very interesting the implementation of water stable isotopes in a climate model as it will offer wide prospects for simulations of past climate. The manuscript reads well overall, some figure captions or figures could be improved. I have several concerns that I would like to see addressed before the publication of the manuscript. I therefore recommend major revision of the present manuscript. Below are my specific comments.

Specific comments "Part2.1 Ocean model in lines 28-30" Can the authors explained in more details the rescaling of the vertical coordinate?

"Part 2.2 in lines 27, Note that the prescribed atmospheric forcing fields obtained from

the PI ocean state estimate by Kurahashi-Nakamura et al., (submitted) and the corresponding isotopic fluxes are not entirely consistent and might introduce an error in our model simulation". The authors refer to unpublished results here. They should show some results that indicate what could be the error and if the use of ratio of the isotopic content indeed minimize the uncertainty.

"Part 2.3.1 in line 30". The authors compare long-term mean monthly value with GISS sample. This is indeed better than to compare with the annual mean isotope values. However, in the rest of the text it is difficult to know when the comparison is based on monthly value or annual. This is also not clear in the different figures and captions on the manuscript.

"part 3.2 line21" The number of measurements for dD is rather small. According to the GISS database there is more than 1000 data points in dD. This is indeed more reduce than for the d18Osw but enough to realize a data-model comparison. Rather, the authors can mentioned that they choose to focus on the d18O and will work on the dD in the future.

"Figure 3" Is it the annual or monthly value that are plotted for the model? Is it the surface data that are compared to the average 50m of the model or the data between 0 and 50 m? What could be the error associated if this is the surface data versus the average 50m?

"part 3.2 lines 3-4, the subtropical gyres are less enriched...." There is also a discrepancy for the Mediterranean Sea. What is the reason for such discrepancy in the subtropical gyres and Mediterranean region?

"Figures 7 and 8" What is the depth used in the model (50 m?), is it annual or monthly?

"Part 4.1 lines 20 to 30 and Figure 9" A zoom on the artic region would be very helpful here. The isotopic values for rivers discussed in the text could eventually be added to this figure of the artic region.

"Part 4.3: Planktonic foraminiferal d18Oc" When reading part 4.3 it seems that the main discrepancy between data and model results is because of the gametogenic calcification of foraminifera and so that paleotemperature equations derived from plankton-tow data are more appropriate to reconstruct surface water conditions that the commonly used paleotemperature equations like Shackleton (1974) or Kim and O'Neil (1997). This discussion is extremely interesting for paleoceanographic studies. Nonetheless I find that all the potential factors that can affect the d18Oc and so the data-model comparison and mismatch are not developed enough. Indeed, the temperature bias in the model (2°C or more in some regions, see figure 1) can affect significantly the d18Oc reconstruction with the model. Similarly, the bias in d18Osw could contribute significantly to this "biased towards lower values". For example, the d18Osw is 0.4‰ too depleted in the model in comparison to data in the tropics (see part 3.2) and 0.9‰ too enriched in the Arctic Ocean (see part 3.2). These biases can affect the d18Ocalcite reconstruction and comparison. Also, it seems that the shift on figure 8a is more important for tropical species than for polar species. The data-model agreement or disagreement seems different depending the oceanic region (or species considered). So I recommend to the authors to realize a data-model comparison for the d18Oc for the different species of foraminifera separately. This analyze is important not only to try to discuss the oceanic region separately but also because other factors can affect each species of foraminifera in a different way. The seasonality is one of this important factor. Although there is one sentence in the part 4.3 that mention that "seasonality could be a problem and is not considered" it would be interesting to estimate how much bias could be introduce by such inconsideration. One way to do that could be to calculate the simulated seasonal amplitude for ocean calcite d18O in the model. It could be that the "biased towards lower values" is partly or totally explained by a distortion of the foraminifera flux towards a specific season or period than the annual mean. Similarly, the effect of the vertical migration is not completely developed. The author discuss the gametogenic calcification that is indeed related to this effect of vertical migration but the different species that are grouped on Figure 8 have different depth habitats and

this affect their d18Oc. They can also change their depth habitat (for example during upwelling conditions). Again a data-model comparison for each species separately and with a different mean depth habitat of calcification would be interesting. The data on figure 8 are only presented for the first 50 m (although not clearly indicated in the text or on the Figure 8 caption). Although it will be difficult to examine the results for the very surface only (because of the grid of the model), the authors can investigate how the integration of the results for deeper water depth affect the data-model comparison.

The authors also suggest that the more enriched d18Oc values obtained with the equation of Shackleton (1974) is because this equation is based on Uvigerina spp shells that are relatively enriched in 18O. In fact, Shackleton (1974) proposed that Uvigerina peregrina is in isotopic equilibrium with seawater contrary to Cibicides. On the contrary, Bemis et al. (1998) (not cited in the discussion) suggested that Cibicides might also calcify in isotopic equilibrium and that the heavier $\delta$18O values of Uvigerina are due to calcification at lower porewater pH. More recently, Marchitto et al., 2014 (also not cited in the discussion) investigated this difference in more details. Their results agree with Bemis et al. (1998) that Cibicidoides and Planulina appear to be closer to isotopic equilibrium (as represented by the Kim and O'Neil (1997) inorganic precipitates, which is also a matter of debate) than Uvigerina, although scatter in the measurements limits their confidence in this statement. They also recommend that Uvigerina $\delta$18O be adjusted to the Cibicidoides scale by subtracting 0.47‰ and not 0.64‰.They were also unable to discern an impact of bottom water pH on benthic foraminiferal $\delta$18O, but they speculate that Uvigerina's deviation from equilibrium could be explained by admixture of rapidly-precipitated non-equilibrium CaCO3 that would be subject to a pH influence. So, to my knowledge, the question as to why the $\delta$18O of Uvigerina and Cibicides are different remains. The question of the pH influence is also not discussed for planktonic foraminifera whereas it could also have a significant effect on the oxygen isotopic composition (Bijma et al., 1999; Zeebe 1999).This pH effect could be a primary mechanism to explain the differences between the equations (Mulitza et al., 2004). Again, the pH effect will be different with the latitudes and so it is important to discuss the species

(that are associated to different oceanic regions) separately.

To resume, I like the discussion in part 4.3, this is of strong interest for paleoceano-graphic studies and the gametogenic calcification is a factor that certainly need to be considered. Nonetheless, the authors do not discuss in details all the factors and bi-ases that can affect the d18Oc of their data-model comparison. For each foraminifera specie, how the bias in d18Osw in the model, the depth use in the model to gener-ate the d18Oc signal, the seasonality and vertical migration and the pH can affect the d18Oc signal modelled and the comparison with data? At the end, if we consider all these factors and potential biases for d18Oc and the data-model comparison, can the authors really conclude that the differences between data and model is mainly linked to gametogenic calcification? If the authors cannot confirm their hypothesis in a revised version, they should also reformulate this conclusion from the abstract and conclusion part.

References: Bemis, B. E., Spero, H. J., Bijma, J., & Lea, D. W. (1998). Reevaluation of the oxygen isotopic composition of planktonic foraminifera: Experimental results and revised paleotemperature equations. Paleoceanography, 13(2), 150-160. Bijma, J., Spero, H. J., & Lea, D. W. (1999). Reassessing foraminiferal stable isotope geochem-istry: Impact of the oceanic carbonate system (experimental results). In Use of proxies in paleoceanography (pp. 489-512). Springer Berlin Heidelberg.

Kim, S. T., & O'Neil, J. R. (1997). Equilibrium and nonequilibrium oxygen isotope effects in synthetic carbonates. Geochimica et Cosmochimica Acta, 61(16), 3461-3475.

Marchitto, T. M., Curry, W. B., Lynch-Stieglitz, J., Bryan, S. P., Cobb, K. M., & Lund, D. C. (2014). Improved oxygen isotope temperature calibrations for cosmopolitan benthic foraminifera. Geochimica et Cosmochimica Acta, 130, 1-11.

Mulitza, S., Donner, B., Fischer, G., Paul, A., Pätzold, J., Rühlemann, C., and Segl, M.: The South Atlantic oxygen-isotope record of planktic foraminifera, in: The South

Atlantic in the Late Quaternary: Reconstruction of Mass Budget and Current Systems edited by Fischer, G. and Wefer, G., 121-142, Springer, New York, 2004.

Shackleton, N. J.: Attainment of isotopic equilibrium between ocean water and the benthonic foraminifera genus Uvigerina: isotopic changes in the ocean during the last glacial, Gif-sur-Yvette, Colloque international du CNRS, 219, 203-210, 1974.

Zeebe, R. E. (1999). An explanation of the effect of seawater carbonate concentration on foraminiferal oxygen isotopes. Geochimica et Cosmochimica Acta, 63(13), 2001-2007.

---

## Referee Comment (RC2) · Anonymous Referee #2 · 21 Mar 2017

The paper Âń stable water isotopes in the MITgcm Âż from Völpel et al, is presenting the implementation of stable water isotopes in the MIT ocean general circulation model. The simulation is evaluated by comparison with seawater oxygen stable isotopes observations and data from planktonic foraminifera. This approach represents an unavoidable stage before using this proxy for assessing past-climate simulations. The manuscript describes the methodology adopted in their modelling approach, but some points are still confusing or not well defined. The analysis is too superficial to correctly assess the performance of the model. I then recommend major revision before publication.

Page 2, line 12. Many references are missing for stable water isotopes in oceanic models: for instance, Delaygue et al, 2000, Roche et al, 2004, etc.. Page2, Line 23. Please define "checkpoint 64w" Page 3 and 4, section 2.2 : This section describes the methodology used for implementing the water isotopes in the MITGcm model. The

simulation is forced with isotopic quantities derived from the NCAR IsoCam model , isotopic composition in precipitation and water vapor for evaporation. These quantities should be presented and discussed at least briefly in the manuscript, in order to allow further discussing the impact on oceanic model performance. Futhermore, it is also useful as the same isotopic oceanic MITGcm model could further be used with other atmospheric model forcing, it will offer the possibility to compare with the results from this study. Page 4, Line 24: value of Ce is not specified

Page 5; line 3: the presentation of the architecture of the code should be more explicit: Apparently, gchem represent the "source and sinks" module and "ptracers" the transport module. Page 5,line 7: Fw should be expilicitely defined as Evap - precip –Runoff

Page 5 line 17. The freshwater flux is balanced by adjusting the precipitation field (page 3 line 14). The adjustment applied to water isotopes simulation must be described and a discussion on how it can potentially affect 18O-Salinity relation is necessary.

Page 5: what is the duration of the spin-up of the simulation?

Page 6 – results Section 3.1 presents model performance for temperature and salinity. Salinity anomaly should be analyzed considering the characterisctics of Evaporation and Precipitation forcing fields used in this study. This will also be useful for next analyzing the water isotopes simulations.

Page 6: figure 3 : color scale is not adapted. Range (-1, 1°/oo) is too narrow to represent the more elevated values of the observations.

Page 6and 7 and discussion: discussion of water isotope distribution in ocean water: the discussion is too superficial. Shortcomings in water mass isotopic composition is described but the causes are never analyzed in function of model dynamical performances (AABW, NADW, AAIW formation) or surface boundary conditions (precipitation, evaporation, isotopic composition in precipitation and water vapor). A minimum

more detailed analysis of the simulation is required to assess this modelling approach.

Pages 9-10: discussion - the discussion of the sources of errors are mainly focuses on rivers input and sea-ice melting for the Arctic Ocean. The discussion must also consider more quantitatively the shortcomings associated to surface boundary conditions (for instance, an analysis of the realism of the isotopic composition in precipitation of the forcing has to be presented and considered, see previous comment).

Page 11- discussion Planktonic Foraminiferal: Observation is a compilation of isotopic measurements derived on different species. The isotopic signal they register is then not obvious since the different species are living at different depth and differently affected by the seasonal cycle. A more sophisticated approach, taking into account the characteristics of some species would be more appropriate.

---

## Referee Comment (RC3) · Anonymous Referee #3 · 23 Mar 2017

Manuscript:
Stable water isotopes in the MITgcm (checkpoint 64w)

Authors:
Rike Völpel, Andre Paul, Annegret Krandick, Stefan Mulitza and Michael Schulz

Journal:
Geoscientific Model Development

Review

Völpel et al. present in their manuscript first results of a newly implemented stable water isotope (SWI) diagnostics within the ocean GCM MITgcm. Their evaluation of this model enhancement focuses on modelling results of $H_2^{18}O$ in a simulation under pre-industrial climate conditions. This evaluation contains both a model-data comparison using measurements of $\delta^{18}O$ in seawater and in planktic foraminifera as well as a brief analysis of the simulated $\delta^{18}O$-salinity relationship in different water bodies.

The manuscript is well outlaid and written in a clear and concise manner. The implementation of SWI into the MITgcm does not contain any new methodological approaches or intellectual merit. It follows more or less directly previous isotope implementations done in other ocean GCM. However, given the few number of existing ocean GCM with SWI diagnostics so far, I still rate this work as highly valuable and well suited for publication in GMD.

Two important issues should be addressed by the authors before publication can be warranted:

(i) It is stated that both stable isotopes $H_2^{18}O$ and HDO have been implemented into MITgcm. However, neither simulated HDO nor Deuterium excess values are discussed anywhere in the manuscript. Even if the number of available $\delta D$ in seawater observations (e.g. GISS database, Schmidt et al., 1999) or comparable model results (e.g. Xu et al., 2012) are limited, a first-order comparison would still be valuable and of high interest for on-going SWI modelling efforts within the scientific community.
Alternatively, the authors might justify in more detail why they have included HDO in the MITgcm, but don't present any of the results in their paper.

(ii) In the manuscript, the printed equation for the equilibrium fraction factor $\alpha_{l-v}$ for HDO is wrong. In Eq. 7 of the manuscript, $\alpha_{l-v}$ is calculated as:

$$\alpha_{l-v} = \exp\left(28.844/SST^2 * 10^3 - 76.248/SST - 5.2612*10^{-2}\right)$$

The correct equation (see Majoube, 1971, Eq. 2) reads:

$$\alpha_{l\text{-}v} = \exp(24.844/SST^2 * 10^3 - 76.248/SST + 5.2612*10^{-2})$$

As no HDO results are shown in this study, I cannot say if this error is simply a (double) typo in the manuscript or if the authors have indeed used a wrong HDO equilibrium factor $\alpha_{l\text{-}v}$ in their simulations. In any case, this severe error has to be checked and corrected before publication.

Further comments and corrections:

- Title: I suggest dropping the information "(checkpoint 64w)" from the title. It is sufficient mentioning the specific MITgcm model release in the Methods section.
- P2, L4/5 (=page 2, line 4/5): I recommend adding some more key references about the application of SWI in ice cores and speleothems. Just citing the studies by Johnsen et al., 2001, and Fleitmann et al., 2003, seems odd and arbitrary.
- P2, L9: correct "form" => "from"
- P2, L16: please explain in more detail why a non-linear free-surface is essential to simulate the δ-salinity relationship properly.
- P2, L26: the chosen vertical model resolution (15 levels) appears to be rather coarse. Please briefly discuss how this might affect the SWI simulation and model-data comparison.
- P3, L16-18: what has been the exact criteria to determine if the PI simulation has reached "quasi steady-state"? Do SWI trends in deep ocean waters still exist at the end of the final 3000 simulation years?
- P3, L25-27: why have different PI atmospheric forcing fields and isotopic fluxes been used for this simulation setup? Wouldn't it have been much more consistent to take all necessary forcing fields from the Tharammal et al., 2013, IsoCAM simulation?
- P4, Eq 11: why is river runoff $R^i$ subtracted in this equation? Conventionally, it is added to $(P^i\text{-}E^i)$ to calculate the total isotopic surface flux.
- P5, L17/18: please quantify the applied correction factor for SWI precipitation. How fast and how much would the global SWI concentration in the ocean change without this correction factor?
- P7, L20/Fig. 4b: if the authors rate the $\delta^{18}O_w$ measurements from the Okhotsk Sea as not representative for the North Pacific, these data points should be omitted in the analyses as well as Fig 4b.
- P7, L24-26: please specify the sample number N for the different correlation calculations.
- P8, L7: omit "nicely"
- P8, L10: correct(?) "Simulated surface waters" => "Simulated calcite values"

- P9, L2: replace "is overestimated" by "is too depleted"
- P9, L2: please add "all other three Russian rivers"
- P9, 10-12: how well do the simulated annual discharge amounts agree with the observational data given in Cooper et al. (2008)? For a correct simulation of river runoff SWI into the ocean, both δ-values and total water amount are of importance.
- P10, 27-29: I don't fully understand this argument. Please explain in some more detail the linkage between salinity restoring and SWI modelling.
- P11, Section 4.3: as a non-expert on planktonic foraminiferal $\delta^{18}O_c$ data, I am a bit confused by this paragraph and the given recommendations. If it is well known that core-top sediments are enriched in $\delta^{18}O_c$ due to gametogenic calcification, why have these data been compared to the simulated planktonic $\delta^{18}O_c$ values at all? And is the better agreement of sediment $\delta^{18}O_c$ data with modelled $\delta^{18}O_c$ calculated with Shackleton's equation just by chance, then? Which procedure/equation do the authors suggest for future SWI modelling studies, if modelled $\delta^{18}O$ values shall be compared to the manifold of available planktonic $\delta^{18}O_c$ values from marine sediments?
- P12, L15: "using real freshwater and isotopic flux boundary conditions" => omit "real"
- P12, L25: omit "remarkably"
- P19, Table 1: please specify in more detail how the different water masses (AAIW, NADW, AABW) have been defined and how the related $\delta^{18}O$ values have been calculated.
- P21, Fig. 2: add the unit "[psu]" to the colour bar title.
- P23, Fig. 4: please specify in more detail how the zonally averaged cross sections of $\delta^{18}O$ have been calculated.
- P23, Fig. 4: why do the plots stop at 50°S and 50°N, respectively? GISS data from higher latitudes exist and it would be valuable to compare model results and observational data in these regions of the Atlantic and Pacific, too.
- P25, Fig. 6: do the plots show salinity and $\delta^{18}O$ values at a depth of 50m or at a depth range 0-50m? Please clarify this in the figure caption.

---

## Author Comment (AC3) · 24 May 2017

The comment was uploaded in the form of a supplement:
http://www.geosci-model-dev-discuss.net/gmd-2017-7/gmd-2017-7-AC3-supplement.pdf

---

## Author Response (AR1)

Authors response to reviewer #1

Dear Reviewer,

We highly appreciate your time and effort spent on reviewing our manuscript. We have prepared a new version of the
5 manuscript with your comments taken into account. Below we include a point-by-point reply to each comment.

**Comment:**
"Part2.1 Ocean model in lines 28-30" Can the authors explained in more details the rescaling of the vertical coordinate?
**Response:**
10 Done.

**Comment:**
"Part 2.2 in lines 27, Note that the prescribed atmospheric forcing fields obtained from the PI ocean state estimate by
Kurahashi-Nakamura et al., (submitted) and the corresponding isotopic fluxes are not entirely consistent and might introduce
15 an error in our model simulation". The authors refer to unpublished results here. They should show some results that indicate
what could be the error and if the use of ratio of the isotopic content indeed minimize the uncertainty.
**Response:**
The paper by Kurahashi-Nakamura et al., (2017) has now been published, so we added the final citation. Using the ratio of the
isotopic content of precipitation and water vapor inevitably leads to isotopic fluxes that are consistent with the optimized
20 precipitation field.  To show some results, we would have to run additional simulations.

**Comment:**
"Part 2.3.1 in line 30". The authors compare long-term mean monthly value with GISS sample. This is indeed better than to
compare with the annual mean isotope values. However, in the rest of the text it is difficult to know when the comparison is
25 based on monthly value or annual. This is also not clear in the different figures and captions on the manuscript.
**Response:**
We changed both figure captions and parts of the text, to make it clearer whether we used annual mean or long-term monthly
mean isotope values for the comparison.

30 **Comment:**
"Part 3.2 line21" The number of measurements for dD is rather small. According to the GISS database there is more than 1000
data points in dD. This is indeed more reduce than for the d18Osw but enough to realize a data- model comparison. Rather,
the authors can mentioned that they choose to focus on the d18O and will work on the dD in the future.
**Response:**

We agree and rephrased the respective sentence. Due to the higher number of measurements we chose to focus on $\delta^{18}O$ to validate our model. However, since $\delta D$ is now also used as a proxy in marine archives (i.e. Häggi et al. 2016) an implementation at this stage seemed reasonable.

5 **Comment:**
"Figure 3" Is it the annual or monthly value that are plotted for the model? Is it the surface data that are compared to the average 50m of the model or the data between 0 and 50 m? What could be the error associated if this is the surface data versus the average 50m?

**Response:**
10 The figure (now Fig. 4) shows the global annual mean surface (upper 50 m) $\delta^{18}O_w$ distribution of the model, while the GISS data are averaged over the upper 50 m. We rephrased the figure caption accordingly. The figure is just a first visualization of the model-data fit. For the statistical comparison (e.g. Fig. 6) we interpolated the GISS samples to the nearest tracer grid point of our model grid using inverse distance weighting. Hence, there is no error associated with the data either if it is surface data or data averaged over the upper 50 m.

**Comment:**
"Part 3.2 lines 3-4, the subtropical gyres are less enriched...." There is also a discrepancy for the Mediterranean Sea. What is the reason for such discrepancy in the subtropical gyres and Mediterranean region?

**Response:**
20 Based on the investigation of simulated E, P and $\delta^{18}O_w$ in P, we can conclude that the discrepancies in the subtropical gyres and the Mediterranean Sea are caused by enhanced P (having a dilutional effect on the surface water) and reduced E, whereby not enough $^{16}O$ is removed from the ocean surface. Even though the comparison with observed $\delta^{18}O_w$ in P is based on rather sparse data, the distribution of $\delta^{18}O_w$ in P seems to be reasonably well simulated. Further, one can assume that $\delta^{18}O_w$ in E is also slightly too enriched, but unfortunately, we cannot confirm this assumption because no observed data exists. We added
25 these assumptions to section 4.2 and 4.3.

**Comment:**
"Figures 7 and 8" What is the depth used in the model (50 m?), is it annual or monthly?

**Response:**
30 The original figures showed the global annual mean $\delta^{18}O_c$ values of the surface (upper 50 m) simulated by the MITgcm. However, in response to the reviews we revised the discussion part (now section 4.4) and thus the figures (now Fig. 8 and 10) changed too. Now, we only compare modeled $\delta^{18}O_c$ values with foraminiferal calcite of plankton tow data (see last response).

Therefore, we interpolated the plankton tow data to the nearest tracer grid point of our model grid using inverse distance weighting and thus compared them to the modeled $\delta^{18}O_c$ values of the respective month and depth level of sampling.

**Comment:**

"Part 4.1 lines 20 to 30 and Figure 9" A zoom on the artic region would be very helpful here. The isotopic values for rivers discussed in the text could eventually be added to this figure of the artic region.

**Response:**

We added a zoom on the Arctic Ocean to the figure (now Fig. 13), including the approximate location of discharge of the six rivers discussed in the text. Furthermore, we included a table (Table 2) to improve the comparison between simulated river values and observed river values by Cooper et al. (2008).

**Comment:**

"Part 4.3: Planktonic foraminiferal d18Oc" When reading part 4.3 it seems that the main discrepancy between data and model results is because of the gametogenic calcification of foraminifera and so that paleotemperature equations derived from plankton-tow data are more appropriate to reconstruct surface water conditions that the commonly used paleotemperature equations like Shackleton (1974) or Kim and O'Neil (1997). This discussion is extremely interesting for paleoceanographic studies. Nonetheless I find that all the potential factors that can affect the d18Oc and so the data-model comparison and mismatch are not developed enough. Indeed, the temperature bias in the model (2° C or more in some regions, see figure 1) can affect significantly the d18Oc reconstruction with the model. Similarly, the bias in d18Osw could contribute significantly to this "biased towards lower values". For example, the d18Osw is 0.4‰ too depleted in the model in comparison to data in the tropics (see part 3.2) and 0.9‰ too enriched in the Arctic Ocean (see part 3.2). These biases can affect the d18Ocalcite reconstruction and comparison. Also, it seems that the shift on figure 8a is more important for tropical species than for polar species. The data-model agreement or disagreement seems different depending the oceanic region (or species considered). So I recommend to the authors to realize a data-model comparison for the d18Oc for the different species of foraminifera separately. This analyze is important not only to try to discuss the oceanic region separately but also because other factors can affect each species of foraminifera in a different way. The seasonality is one of this important factor. Although there is one sentence in the part 4.3 that mention that "seasonality could be a problem and is not considered" it would be interesting to estimate how much bias could be introduce by such inconsideration. One way to do that could be to calculate the simulated seasonal amplitude for ocean calcite d18O in the model. It could be that the "biased towards lower values" is partly or totally explained by a distortion of the foraminifera flux towards a specific season or period than the annual mean. Similarly, the effect of the vertical migration is not completely developed. The author discuss the gametogenic calcification that is indeed related to this effect of vertical migration but the different species that are grouped on Figure 8 have different depth habitats and this affect their d18Oc. They can also change their depth habitat (for example during upwelling conditions). Again a data-model comparison for each species separately and with a different mean depth habitat of calcification would be interesting.

The data on figure 8 are only presented for the first 50 m (although not clearly indicated in the text or on the Figure 8 caption). Although it will be difficult to examine the results for the very surface only (because of the grid of the model), the authors can investigate how the integration of the results for deeper water depth affect the data-model comparison.

The authors also suggest that the more enriched d18Oc values obtained with the equation of Shackleton (1974) is because this equation is based on Uvigerina spp shells that are relatively enriched in 18O. In fact, Shackleton (1974) proposed that Uvigerina peregrina is in isotopic equilibrium with seawater contrary to Cibicides. On the contrary, Bemis et al. (1998) (not cited in the discussion) suggested that Cibicides might also calcify in isotopic equilibrium and that the heavier $\delta18O$ values of Uvigerina are due to calcification at lower porewater pH. More recently, Marchitto et al., 2014 (also not cited in the discussion) investigated this difference in more details. Their results agree with Bemis et al. (1998) that Cibicidoides and Planulina appear to be closer to isotopic equilibrium (as represented by the Kim and O'Neil (1997) inorganic precipitates, which is also a matter of debate) than Uvigerina, although scatter in the measurements limits their confidence in this statement. They also recommend that Uvigerina $\delta18O$ be adjusted to the Cibicidoides scale by subtracting 0.47‰ and not 0.64‰. They were also unable to discern an impact of bottom water pH on benthic foraminiferal $\delta18O$, but they speculate that Uvigerina's deviation from equilibrium could be explained by admixture of rapidly-precipitated non-equilibrium CaCO3 that would be subject to a pH influence. So, to my knowledge, the question as to why the $\delta18O$ of Uvigerina and Cibicides are different remains. The question of the pH influence is also not discussed for planktonic foraminifera whereas it could also have a significant effect on the oxygen isotopic composition (Bijma et al., 1999; Zeebe 1999). This pH effect could be a primary mechanism to explain the differences between the equations (Mulitza et al., 2004). Again, the pH effect will be different with the latitudes and so it is important to discuss the species (that are associated to different oceanic regions) separately.

To resume, I like the discussion in part 4.3, this is of strong interest for paleoceanographic studies and the gametogenic calcification is a factor that certainly need to be considered. Nonetheless, the authors do not discuss in details all the factors and biases that can affect the d18Oc of their data-model comparison. For each foraminifera specie, how the bias in d18Osw in the model, the depth use in the model to generate the d18Oc signal, the seasonality and vertical migration and the pH can affect the d18Oc signal modelled and the comparison with data? At the end, if we consider all these factors and potential biases for d18Oc and the data-model comparison, can the authors really conclude that the differences between data and model is mainly linked to gametogenic calcification? If the authors cannot confirm their hypothesis in a revised version, they should also reformulate this conclusion from the abstract and conclusion part.

**Response:**

We agree with reviewer 1 that our discussion of foraminiferal $\delta^{18}O_c$ was too far-reaching. Indeed, many other processes (i.e. seasonal and vertical calcification, dissolution) exist that influence the composition of foraminiferal shells recorded at the sea floor, besides $\delta^{18}O_w$ and temperature. Since our model does not have an ecosystem module, many of these processes are not simulated, and we feel that our model is not the right tool to either gain information on foraminiferal ecology or on model performance. For this reason, we refrain from comparing our model results to core top $\delta^{18}O_c$ in the revised version of the paper. Plankton tow data are better constrained with respect to the time and depth of calcification. In order to demonstrate how the

combined simulation of seawater temperature and $\delta^{18}O_w$ reflects the isotopic composition of foraminiferal carbonate, we hence kept the comparison to plankton tow data.

Authors response to reviewer #2

Dear Reviewer,

We highly appreciate your time and effort spent on reviewing our manuscript. We have prepared a new version of the
5  manuscript with your comments taken into account. Below we include a point-by-point reply to each comment.

**Comment:**

Page 2, line 12. Many references are missing for stable water isotopes in oceanic models: for instance, Delaygue et al, 2000,
Roche et al, 2004, etc.

10  **Response:**

Done.

**Comment:**

Page2, Line 23. Please define "checkpoint 64w"

15  **Response:**

Done.

**Comment:**

Page 3 and 4, section 2.2: This section describes the methodology used for implementing the water isotopes in the MITgcm
20  model. The simulation is forced with isotopic quantities derived from the NCAR IsoCam model, isotopic composition in
precipitation and water vapor for evaporation. These quantities should be presented and discussed at least briefly in the
manuscript, in order to allow further discussing the impact on oceanic model performance. Furthermore, it is also useful as
the same isotopic oceanic MITgcm model could further be used with other atmospheric model forcing, it will offer the
possibility to compare with the results from this study.

25  **Response:**

We included a comparison of $\delta^{18}O_w$ in P in the section 4.2 and added the global distribution as well as model-data fit as an
additional figure to the manuscript (Fig. 11). Unfortunately, we do not know of any comprehensive compilation of the isotopic
composition in water vapor over the ocean and thus cannot present this field.

30  **Comment:**

Page 4, Line 24: value of Ce is not specified.

**Response:**

$C_E$ is specified as transfer coefficient for evaporation on page 4, line 30. For the exact definition, we added Large and Yeager
(2004) as a reference, since the calculation of evaporation follows the bulk forcing approach by them.

**Comment:**

Page 5; line 3: the presentation of the architecture of the code should be more explicit: Apparently, gchem represent the "source and sinks" module and "ptracers" the transport module.

**Response:**

To clarify the purposes of the respective packages involved in the simulation of the stable water isotopes, we added an additional Table (Table 1) as an overview and rephrased the respective sentence in section 2.2.

**Comment:**

Page 5, line 7: Fw should be explicitly defined as Evap – precip – Runoff.

**Response:**

Done.

**Comment:**

Page 5 line 17. The freshwater flux is balanced by adjusting the precipitation field (page 3 line 14). The adjustment applied to water isotopes simulation must be described and a discussion on how it can potentially affect 18O-Salinity relation is necessary.

**Response:**

We added the description of the calculation of the tracer specific correction factor to Appendix A and also modified Fig. A1. Due to the correction factors both the global salinity and $\delta^{18}O_w$ remains constant. Thus, any artificial drifts are prevented, which would otherwise lead to inexplicable changes in the y-intercept and slope of the $\delta^{18}O_w$-salinity relationship.

**Comment:**

Page 5: what is the duration of the spin-up of the simulation?

**Response:**

The duration of spin-up was 3000 model years (cf. section 2.1, page 3, line 20).

**Comment:**

Page 6 – results Section 3.1 presents model performance for temperature and salinity. Salinity anomaly should be analyzed considering the characteristics of Evaporation and Precipitation forcing fields used in this study. This will also be useful for next analyzing the water isotopes simulations.

**Response:**

For the presentation of the general model performance, we added zonally-averaged cross sections of temperature and salinity through the Atlantic Ocean and compared them to the GISS data. Regarding the salinity, we present precipitation and evaporation anomalies in section 4.1.

**Comment:**

Page 6: figure 3 : color scale is not adapted. Range (-1, 1 ‰) is too narrow to represent the more elevated values of the observations.

**Response:**

Done. (now Fig. 4)

**Comment:**

Page 6and 7 and discussion: discussion of water isotope distribution in ocean water: the discussion is too superficial. Shortcomings in water mass isotopic composition is described but the causes are never analyzed in function of model dynamical performances (AABW, NADW, AAIW formation) or surface boundary conditions (precipitation, evaporation, isotopic composition in precipitation and water vapor). A minimum more detailed analysis of the simulation is required to assess this modelling approach.

**Response:**

As described above, we added zonally-averaged cross sections of temperature and salinity through the Atlantic Ocean and compared them to the GISS data. Further we present precipitation and evaporation anomalies in section 4.1. The general model performance is shortly discussed in section 4.1, while the isotopic composition in precipitation is presented and discussed in section 4.2.

**Comment:**

Pages 9-10: discussion - the discussion of the sources of errors are mainly focuses on rivers input and sea-ice melting for the Arctic Ocean. The discussion must also consider more quantitatively the shortcomings associated to surface boundary conditions (for instance, an analysis of the realism of the isotopic composition in precipitation of the forcing has to be presented and considered, see previous comment).

**Response:**

We expanded the discussion on the sources of error by e.g. considering the surface boundary conditions.

**Comment:**

Page 11- discussion Planktonic Foraminiferal: Observation is a compilation of isotopic measurements derived on different species. The isotopic signal they register is then not obvious since the different species are living at different depth and

differently affected by the seasonal cycle. A more sophisticated approach, taking into account the characteristics of some species would be more appropriate.

**Response:**

We changed our comparison of modeled $\delta^{18}O_c$ with measurements by using only plankton-tow data. Since the isotopic composition of the foraminiferal shell may be altered by mechanisms such as vital effects, vertical migration and modifications after death, a comparison with living foraminifera, where the depth and month of sampling in known, seemed to be more appropriate for testing the capability of the model on reconstructing $\delta^{18}O_c$. This way different depth habitats for the different species as well as seasonal peaks should be overcome. Further, we also performed a model-data comparison for each species separately to get a better idea on sources of error for the $\delta^{18}O_c$.

Authors response to reviewer #3

Dear Reviewer,

We highly appreciate your time and effort spent on reviewing our manuscript. We have prepared a new version of the
manuscript with your comments taken into account. Below we include a point-by-point reply to each comment.

**Comment:**

It is stated that both stable isotopes H218O and HDO have been implemented into MITgcm. However, neither simulated HDO
nor Deuterium excess values are discussed anywhere in the manuscript. Even if the number of available δD in seawater
observations (e.g. GISS database, Schmidt et al., 1999) or comparable model results (e.g. Xu et al., 2012) are limited, a first-
order comparison would still be valuable and of high interest for on-going SWI modelling efforts within the scientific
community. Alternatively, the authors might justify in more detail why they have included HDO in the MITgcm, but don't
present any of the results in their paper.

**Response:**

For our main goal in the near future we want to use the stable water isotope package for paleoclimatic reconstructions and
compare those simulations with available $\delta^{18}O_c$ of mainly benthic foraminifera. Thus, it made more sense to us to expound our
model validation on the $\delta^{18}O_w$ distribution, since this is one of the main factors influencing the oxygen isotopic composition
of foraminiferal shells. We slightly rephrased the beginning of part 3.2 to indicate the latter. Nevertheless, we wanted to
implement HDO as a passive tracer as well to simplify the comparison with other models during investigations in the future
either by the authors or other researchers who would like to use the newly developed package.

**Comment:**

In the manuscript, the printed equation for the equilibrium fraction factor $\alpha_{l-v}$ for HDO is wrong. In Eq. 7 of the manuscript,
$\alpha_{l-v}$ is calculated as:

$\alpha_{l-v} = \exp(28.844/SST^2 * 10^3 - 76.248/SST - 5.2612*10^{-2})$

The correct equation (see Majoube, 1971, Eq. 2) reads:

$\alpha_{l-v} = \exp(28.844/SST^2 * 10^3 - 76.248/SST + 5.2612*10^{-2})$

As no HDO results are shown in this study, I cannot say if this error is simply a (double) typo in the manuscript or if the authors
have indeed used a wrong HDO equilibrium factor $\alpha_{l-v}$ in their simulations. In any case, this severe error has to be checked
and corrected before publication.

**Response:**

Unfortunately, this was a typo in the manuscript. We corrected Eq. 2.

**Comment:**

Title: I suggest dropping the information " (checkpoint 64w) " from the title. It is sufficient mentioning the specific MITgcm model release in the Methods section.
**Response:**
Done.

**Comment:**
P2, L4/5 (=page 2, line 4/5): I recommend adding some more key references about the application of SWI in ice cores and speleothems. Just citing the studies by Johnsen et al., 2001, and Fleitmann et al., 2003, seems odd and arbitrary.
**Response:**
10   Done.

**Comment:**
P2, L9: correct "form" => "from"
**Response:**
15   Done.

**Comment:**
P2, L16: please explain in more detail why a non-linear free-surface is essential to simulate the δ-salinity relationship properly.
**Response:**
20   The salinity in the global ocean changes due to freshwater fluxes through the air-sea interface altering the ocean volume. Many ocean as well as coupled atmosphere-ocean models still use a virtual salt flux to mimic this effect. In this case, the conservation of salt requires a constant reference salinity (usually taken to be the global annual-mean surface salinity) to estimate the virtual salt flux. This way, regions that differ significantly from the reference value might be biased and the SSS cannot evolve freely. Furthermore, a restoring formulation is often employed by adding a relaxation term to the virtual salt flux to reproduce an SSS
25   distribution that fits the observations (Huang, 1993; Roullet and Madec, 2000). This approach gives no insight in the dynamical explanation of the SSS pattern and is particularly problematic for past or future climate conditions, for which the SSS is unknown.
Using the real freshwater flux boundary conditions, the concentration and/or dilution effect is accurately simulated, whereby ocean volume changes and the SSS evolves freely. In the case of an ocean model with a free surface (as opposed to a rigid lid),
30   a fully non-linear formulation of the free surface is required to conserve global ocean salinity.
The same reasoning applies to global $\delta^{18}O_w$ as well. Due to the real freshwater flux boundary conditions in conjunction with the non-linear free surface the salinity and $\delta^{18}O_w$ are dynamically more accurately simulated and thus the resulting $\delta^{18}O$-salinity relationship is expected to be much more realistic.
We added this information in condensed from to the introduction.

**Comment:**

P2, L26: the chosen vertical model resolution (15 levels) appears to be rather coarse. Please briefly discuss how this might affect the SWI simulation and model-data comparison.

5  **Response:**

Since the upper ~500 m are presented by only 4 layers in the ocean model, the thermocline might not be as pronounced as in the real ocean. Observational data that corresponds to depths within this transitional layer might reflect a different signal than resolved by the ocean model. More generally, a coarse vertical resolution makes a realistic representation of water mass boundaries and the comparison to observations (which involves vertical interpolation) more difficult. We added this

10  information to section 4.2.

**Comment:**

P3, L16-18: what has been the exact criteria to determine if the PI simulation has reached "quasi steady-state"? Do SWI trends in deep ocean waters still exist at the end of the final 3000 simulation years?

15  **Response:**

We determined the quasi stead-state based on the salinity, temperature and Atlantic Meridional Overturning Circulation (AMOC). There were no critical trends visible, so that the global salinity, temperature and AMOC were approximately steady at 34.73 psu, 2.86° C and 18.24 Sv after 3000 model years, respectively. We added these criteria to the manuscript.

Regarding the stable water isotopes in the global ocean, there are no observable trends visible after the final 3000 years (for

20  more details see response to comment to "correction factor for stable water isotope precipitation"), thus the global tracer concentration in the ocean was conserved (cf. P5, L21-22).

**Comment:**

P3, L25-27: why have different PI atmospheric forcing fields and isotopic fluxes been used for this simulation setup? Wouldn't

25  it have been much more consistent to take all necessary forcing fields from the Tharammal et al., 2013, IsoCAM simulation?

**Response:**

Yes, it would have been, but unfortunately, we were not able to directly force the MITgcm with these fields because they led to a collapse of the overturning circulation. Therefore, we decided to use the forcing fields by Kurahashi-Nakamura et al., (2017), which were optimized to produce proper hydrographic conditions in our configuration of the MITgcm.

**Comment:**

P4, Eq 11: why is river runoff Ri subtracted in this equation? Conventionally, it is added to ($P^i$-$E^i$) to calculate the total isotopic surface flux.

**Response:**

Thank you for pointing that out. In Eq. 11 a bracket is missing. We changed Eq. 11 to:

$$F^i = -((E^i - P^i) \cdot (1 - A_{ice}) - R^i)$$

**Comment:**

P5, L17/18: please quantify the applied correction factor for SWI precipitation. How fast and how much would the global SWI concentration in the ocean change without this correction factor?

**Response:**

Without the correction factor, the stable water isotopes would not have reached a "quasi steady-state". Even after 3000 model years a continuous trend in the concentration of stable water isotopes would exist, resulting in a global $\delta^{18}O_w$ value of -0.17 ‰ (the ocean was initialized with 0 ‰). With the correction factor the stable water isotopes are steady with a global $\delta^{18}O_w$ value of -0.0003 ‰. For the quantification of the applied correction factor we modified Fig. A1 and added some additional information to the appendix A.

**Comment:**

P7, L20/Fig. 4b: if the authors rate the $\delta^{18}Ow$ measurements from the Okhotsk Sea as not representative for the North Pacific, these data points should be omitted in the analyses as well as Fig 4b.

**Response:**

The $\delta^{18}O_w$ measurements from the Okhotsk Sea are not representative for a zonally-averaged cross section of the North Pacific. To point this out we rephrased the respective sentence. However, since we show them in Fig. 4 and also use them in the model-data comparison in Fig. 6 (where we use inverse distance weighting to interpolate the GISS data to our model grid) we would like to keep them in Fig. 5b as well.

**Comment:**

P7, L24-26: please specify the sample number N for the different correlation calculations.

**Response:**

Done.

**Comment:**

P8, L7: omit "nicely"

**Response:**

Done.

**Comment:**

P8, L10: correct(?) "Simulated surface waters" => "Simulated calcite values"

**Response:**

Done.

5   **Comment:**

P9, L2: replace "is overestimated" by "is too depleted"

**Response:**

Done.

10  **Comment:**

P9, L2: please add "all other three Russian rivers"

**Response:**

Done.

15  **Comment:**

P9, 10-12: how well do the simulated annual discharge amounts agree with the observational data given in Cooper et al. (2008)? For a correct simulation of river runoff SWI into the ocean, both δ-values and total water amount are of importance.

**Response:**

It is difficult to determine the annual discharge amount in the MITgcm, because determining the grid cells that belong to the
20  respective river is based on visually assigning them according to the location of the river mouth. This may lead to deviations compared to observational data. Nevertheless, when doing so, we get good agreement for the Yenisey, Lena, Yukon and Mackenzie rivers, while the Ob' and Kolyma rivers differ substantially. But if we compare the total modeled discharge received by the Arctic basin (> 60° N) it fits quite well the observational estimate. We added this information to section 4.2 and further included a Table (Table 2) to improve the comparison.

**Comment:**

P10, 27-29: I don't fully understand this argument. Please explain in some more detail the linkage between salinity restoring and SWI modelling.

**Response:**

30  As described in one of the previous responses, salinity restoring gives no insights in the dynamical explanation of the SSS pattern and is problematic for past climate conditions, where SSS is. Investigating the $\delta^{18}$O-salinity requires a free simulation of salinity and $\delta^{18}O_w$, which is possible by combining the nonlinear free surface and a real freshwater flux boundary condition. Thus, our results are an improvement compared to ocean models using virtual salt fluxes and salinity restoring, where a freely evolving salinity is not ensured.

**Comment:**

P11, Section 4.3: as a non-expert on planktonic foraminiferal $\delta^{18}O_c$ data, I am a bit confused by this paragraph and the given recommendations. If it is well known that core-top sediments are enriched in $\delta^{18}O_c$ due to gametogenic calcification, why have these data been compared to the simulated planktonic $\delta^{18}O_c$ values at all? And is the better agreement of sediment $\delta18Oc$ data with modelled $\delta^{18}O_c$ calculated with Shackleton's equation just by chance, then? Which procedure/equation do the authors suggest for future SWI modelling studies, if modelled $\delta^{18}O$ values shall be compared to the manifold of available planktonic $\delta^{18}O_c$ values from marine sediments?

**Response:**

We agree and changed our comparison of modeled $\delta^{18}O_c$ with measurements by using only plankton-tow data. Since $\delta^{18}O_c$ values measured on planktonic foraminifera from sediment cores might not reflect equilibrium with the surface ocean water due to mechanisms such as vital effects, vertical migration and modifications after death, it seemed to be more appropriate testing modeled $\delta^{18}O_c$ against living foraminifera from plankton-tows. This way any deviations due to seasonality, depth habitat, gametogenic calcification or modifications after death should be negligible.

The better agreement using the Shackleton equation was indeed due to the circumstance that it is based on the benthic foraminifera *Uvigerina spp*, which is relatively enriched in $^{18}O$ and produces a similar offset from equilibrium calcite as the gametogenic calcification.

The use of ecosystem models including foraminifera might ultimately provide a better understanding of the factors that determine the recording of oxygen isotopes in foraminiferal shells (Fraile et al. (2008), Lombard et al. (2009), Kretschmer et al. (2016)).

**Comment:**

P12, L15: "using real freshwater and isotopic flux boundary conditions" => omit "real"

**Response:**

We would like to keep it that way, because the term "real freshwater flux boundary conditions" is based on Huang, 1993.

**Comment:**

P12, L25: omit "remarkably"

**Response:**

Done.

**Comment:**

P19, Table 1: please specify in more detail how the different water masses (AAIW, NADW, AABW) have been defined and how the related $\delta^{18}O$ values have been calculated.

**Response:**

Done.

**Comment:**

P21, Fig. 2: add the unit "[psu]" to the colour bar title.

**Response:**

Done.

**Comment:**

P23, Fig. 4: please specify in more detail how the zonally averaged cross sections of $\delta^{18}O$ have been calculated.

**Response:**

For the calculation of the zonally-averaged cross sections we used a mask for the respective basin provided by the WOA09

15 and divided it into equally spaced latitudinal bands. Along those latitudinal bands a weighted zonal mean was calculated. We added those information to the figure caption of the figure (now Fig. 5).

**Comment:**

P23, Fig. 4: why do the plots stop at 50°S and 50°N, respectively? GISS data from higher latitudes exist and it would be

20 valuable to compare model results and observational data in these regions of the Atlantic and Pacific, too.

**Response:**

Our plots in the figure (now Fig. 5) are based on the basin masks provided by the WOA09. According to the WOA09 the Atlantic and Pacific basin extend from 50° S to 60° N, thus we went along with that.

25 **Comment:**

P25, Fig. 6: do the plots show salinity and $\delta^{18}O$ values at a depth of 50m or at a depth range 0-50m? Please clarify this in the figure caption.

**Response:**

The figure (now Fig. 7) shows all the GISS data from a depth range of 0-50 m, either located in the tropics or mid-latitudes,

30 while the data simulated by the MITgcm correspond to the first level of the vertical grid, thus the upper 50 m. We clarified this in the figure caption.

[revised manuscript text omitted]

---

## Author Response (AR2)

Dear Editor and Reviewers,

We would like to thank you for reviewing our revised manuscript. We have prepared a new version of the manuscript with your comment taken into account.

5  **Comment:**

There is only one remaining substantial error that needs to be corrected relating to Equation 7. Please also ensure that it is only a typo and not a more deeply root error.

**Response:**

Done. Unfortunately, we overlooked the typo in the first term of Eq. 7 in the revised manuscript and only corrected the error

10  in the arithmetic operator. Now everything should be correct.

[revised manuscript text omitted]